**Title Page**
**Title:** Simulating growth-based harvest adaptive to future climate change
**List of authors:** Rasoul Yousefpour[1,2*], Julia E.M.S. Nabel[1], Julia Pongratz[1,3]
[1] Land in the Earth System, Max Planck Institute for Meteorology, Bundesstr. 53, 20143
Hamburg, Germany
[2] Chair of Forestry Economics and Forest Planning, University of Freiburg, 79106 Freiburg,
Germany
[3] Now at Department of Geography, Ludwig-Maximilians-Universität, 80333 München,
Germany
**\*Corresponding author:** Rasoul Yousefpour (Rasoul.yousefpour@ife.uni-freiburg.de, Tel:

13 +49-761-2033688)

**Abstract**

Forests are the main source of biomass production from solar energy and take up globally around $2.4 \pm 0.4$ PgC per year. Future changes in climate may affect forest growth and productivity. Currently, state-of-the-art Earth system models use prescribed wood harvest rates in future climate projections. These rates are defined by integrated assessment models (IAMs) only accounting for regional wood demand and largely ignoring the supply side from forests. Therefore, we assess how global growth and harvest potentials of forests change when they are allowed to respond to changes in environmental conditions. For this, we simulate wood harvest rates oriented towards the actual rate of forest growth. Applying this growth-based harvest rule (GB) in "JSBACH", the land component of the Max-Planck-Institute's Earth System Model, forced by several future climate scenarios, we realized a growth potential twice to four times ($3\text{-}9 \text{ PgCy}^{-1}$) the harvest rates prescribed by IAMs ($1\text{-}3 \text{ PgCy}^{-1}$). Limiting GB to managed forest area (MF), we simulated a harvest potential of $3\text{-}7 \text{ PgCy}^{-1}$, two to three times higher than IAMs. This highlights the need to account for the dependence of forest growth on climate. To account for long term effects of wood harvest as integrated in IAMs, we added a life cycle analysis showing that the higher supply with MF as an adaptive forest harvesting rule may improve the net mitigation effects of forest harvest during the $21^{st}$ century by sequestering carbon in anthropogenic wood products.

**Keywords:** Adaptation to climate change, Mitigation, Mortality, Carbon forestry, sustainable forest management, Global forest model

## 1. Introduction

Forest ecosystems play a major role in taking up global $CO_2$ emissions and affect global climate conditions through a range of complex biophysical and biogeochemical processes. Forests are the main source of biomass production from solar energy through photosynthesis and are estimated to take up globally around $2.4 \pm 0.4$ $PgCy^{-1}$ (Pan et al., 2011). A large part of this uptake can be attributed to direct and indirect human interference: Direct human impact by forest management creates young forests sequestering carbon during regrowth (Houghton et al., 2012), and provides material for fossil-fuel substitution (Nabuurs et al., 2007). However, forest utilization and interaction of management with large-scale natural disturbances, such as forest fires, may emit tonnes of $CO_2$ immediately to the atmosphere and act as a source of $CO_2$ emissions (Bonan, 2008). Indirect human impact alters environmental conditions, in particular climate and atmospheric $CO_2$ concentrations, which historically has caused a carbon uptake by the terrestrial vegetation (Le Quéré et al., 2018). Any change in environmental conditions affects forest growth, risks of hazards, and productivity and, consequently, the amount of wood that can be harvested (Temperli et al., 2012; Sohngen and Tian, 2016).

The effects of changes in environmental conditions on the state of the biosphere are represented in state-of-the-art Earth system models (ESMs). However, the description of forest management in these models is largely independent of environmental changes: So far, ESMs employ prescribed wood harvest amounts. These are derived from national statistics for the historical period and from global integrated assessment models (IAMs) for future scenarios. IAMs determine the wood harvest rates based on the supply of woody materials from vegetation and demands of regional industries and population (van Vuuren et al., 2011). However, changes in the supply via forest growth and changed structural conditions especially under climate change and increasing $CO_2$ concentrations are ignored. The main drivers of these models are economic, i.e. market price, and population growth scenarios and forest harvest decisions are only reactive

to the assumed socioeconomic scenarios and do not take forest ecosystem dynamics and growth
into account.
In this study we investigate the relevance of changes in environmental conditions for the growth
potential of forests and subsequently their harvest potentials. Moreover, we explore the
ecological potential of world forest resources for wood production and the implications for
carbon mitigation. To assess growth and harvest potentials, we investigate forest growth under
various future climate scenarios. We allow forests to be harvested and to regrow in response to
the respective changes in environmental conditions, in all scenarios such that the growth
increment is removed each year, i.e. the biomass stocks are neither reduced nor increased. We
call this "growth-based" harvesting (GB). Removing the annual increment mirrors the forest
management concept of "sustained yield". Managing for sustained yield is a strong
sustainability policy applied in sustainable forest management, which aims to maintain forest
stocks as natural capital and controls wood extraction (Luckert and Williamson, 2005).
According to the sustained yield concept, the maximum wood harvest rate to utilize forest
resources equals the actual rate of forest growth. Exceeding regrowth rates would result in the
exploitation of forest ecosystems and would decrease forest yield and productivity. On the other
hand, minimalistic usage, i.e. falling below regrowth rates, would not be an optimal allocation
of forest resources from the perspective of production. However, the traditional concept of
sustained yield management, as defined above, does not account for changes in the growth rates
(Luckert und Williamson, 2005), although forest growth rates are highly dependent on the
environmental conditions (Collins et al., 2018). It has been noted before that any decision about
forest management should take into account the effects of changes in climate and $CO_2$
concentrations on forest growth (Yousefpour et al., 2012; Hickler et al., 2015; Sohngen and
Tian, 2016; Sohngen et al., 2016) and consequently on the harvest rate (Temperli et al., 2012;
Jönsson et al., 2015). Here we demonstrate how altered growth potentials translate into higher
harvest potentials under an adaptive growth-based harvest. We first idealize the concept for this
study in that GB is applied world-wide irrespective of the accessibility of the forest for forest
management activities, but allowing for dependence of wood harvest on altered climate
conditions and $CO_2$ concentrations. In a second, post-processing, step we link these results to
actual harvest potentials by overlaying information on the accessibility of forest areas, where
accessibility refers to any hindrance to use the forest, be it due to conservation or biodiversity
aspects, restricted accessibility due to distance from transport ways or topographical obstacles.
For this step we overlay the map of managed forest area by Kraxner et al. (2017). Kraxner et
al. (2017) used FAO definition for primary forest and as naturally regenerated forest of native
species, where there are no clearly visible indications of human activities and the ecological
processes are not significantly disturbed. Using a map of todays' managed forest as proxy for
accessibility in the future must be seen as a conservative approach, as technological means to
access forests are generally increasing over time.
Keeping in mind the above-mentioned problem of not accounting for changing environmental
conditions in global forest utilization modelling, the goal of this study is to establish a modeling
framework that allows harvesting rates to respond interactively to environmental changes. We
further assess the maximum potential of global forest resources for wood production and the
long-term $CO_2$ mitigation effects of wood harvest, which are implicitly (defining future wood
harvest rate based on RCPs' storyline for mitigation) or explicitly (e.g. using wood for
bioenergy production) the drivers of forest utilization in IAMs. We compare the outcome of the
growth-based harvest with the outcome when applying prescribed wood harvest amounts from
three different Representative Concentration Pathways (RCPs) realized by IAMs and
commonly used by ESMs as an external forcing (Hurtt et al., 2011). Since harvested material
is used in the IAMs to estimate the amount of bioenergy wood, which in turn is needed in the
IAMs to analyze energy and carbon mitigation policies, we perform a first-order assessment of
the $CO_2$ consequences of altering the harvest rates in response to climate. Similarly and to

determine the mitigation potential by wood products we allocate the harvested material to products of different lifetimes according to FAO country-specific statistics (FAOSTAT, 2016). The change in atmospheric carbon content resulting from the release of $CO_2$ by the decay of these products is quantified accounting for compensating fluxes by the ocean and the terrestrial vegetation (Maier-Reimer and Hasselmann, 1987). This impulse response function approach approximates the uptake of emissions by natural sinks in land/ocean, independent of the source, and is a common tool to estimate the fraction of emissions held by the atmosphere over time after the emission occurred (e.g. O'Halloran et al., 2012; Pongratz et al., 2011).

The net mitigation effect of wood harvest is then defined as the difference between the total amount of harvested material and the change in atmospheric carbon content.

2. Materials and methods

2.1. Dynamic global vegetation model JSBACH

We implemented the GB harvesting rule in JSBACH, the land component of the MPI-ESM (Reick et al., 2013). In the applied version of JSBACH vegetation is represented by 12 plant functional types (PFTs) including six woody PFTS. Each PFT is globally endowed with properties in relation to integrated processes in JSBACH and PFT-specific phenology, albedo, morphology, and photosynthetic parameters (Pongratz et al., 2009). Organic carbon is stored in three vegetation pools: living tissue as "green", woody material as "wood", sugar and starches as "reserve pool", and two soil pools with a fast (about 1 year) and a slow (about 100 years) turnover time (Raddatz et al., 2007). Wood harvest activities do not change the area or characteristics of different PFTs, but affect the carbon pools of woody PFTs (forests and shrubs) by removing carbon from the wood pool, resembling trees' stem and branches removal via harvesting (Reick et al., 2013). Harvest thus affects the vegetation carbon stocks, but the model does not represent a feedback of the harvest activity on the forest productivity.

We applied JSBACH in 'offline' mode, i.e. not coupled to the atmosphere, but driven by the
CMIP5 output of the MPI-ESM (Giorgetta et al., 2013) from experiments with $CO_2$ forcing
according to three different RCP scenarios (RCP2.6, RCP4.5 and RCP8.5) for the year 2006-
2100. We used a T63/1.9° horizontal resolution and conducted our simulations with
disturbances due to fire and wind. The simulations were conducted without dynamic vegetation
and without land-use transitions to prevent changes in the areas occupied by the different PFTs
and to be thus able to isolate the effects of forest management activities. Further details on the
model version and the simulation setup are given in the supplementary material (S1).
2.2.RCPs wood harvest
The current standard module for anthropogenic land cover and land-use change in JSBACH is
based on the harmonized land-use protocol (Reick et al., 2013), which provides land-use
scenarios for the period 1500-2100 (Hurtt et al., 2011). As part of this protocol, a set of globally
gridded harvest maps from the IAM implementations of the RCPs is provided (Hurtt et al.,
2011). The prescribed wood harvest rate maps are defined to meet regional timber and
bioenergy demand driven by the increasing population (as the model MESSAGE simulates for
RCP8.5, Riahi et al., 2011), assumptions about population and labor productivity (Model
GCAM for RCP4.5, Brenkert et al., 2003), or demand, trade and supply of agricultural products
and wood based bio-energy (model IMAGE for RCP2.6, van Vuuren et al., 2011). The RCP
wood harvest rates are all based on the demand for wood and bioenergy as the main driver of
decisions by IAMs on forest harvest, neglecting changing availability of forest resources under
environmental changes. For example, RCP8.5 applies the forest sector model DIMA (Riahi et
al., 2011), which is a spatial model for simulating forestry processes to meet specific regional
demand on wood and bioenergy. RCP4.5 bases wood harvest rates solely on the price of carbon
affected by emissions and mitigation potentials of forestry and agricultural activities (Hurtt et
al., 2011). Finally, RCP2.6 relies on the forecasted demand on timber and fuelwood from forest
resources and applies a series of forest management rules (plantation, clear cutting, selective
logging) to meet this demand as the only driver of wood harvest rate in the IMAGE model
(Stehfest et al., 2014). In JSBACH simulations, the harvest prescribed in these maps is fulfilled
taking above-ground carbon of all vegetation pools and all PFTs proportionally to the different
pool sizes. In this study, however, we concentrate on the carbon harvested from the wood pool
of the woody PFTs, which by far contributes most of the harvested volume.
2.3. Growth-based (GB) harvesting rule to estimate growth potentials
As an alternative for the prescribed harvest maps, we implemented the GB harvesting rule,
which allows for adaptive wood harvesting reacting to changes in wood increments, and
accordingly dependent on climate and $CO_2$ conditions. We define the GB rule as the allowance
to harvest specific volumes of wood to the extent of the average increment (i.e. the average
annual growth). Applying GB, we aim to stabilize the wood carbon pool in the woody PFTs at
the level of a selected reference period. In the current paper we selected the maximum level for
the present period (1996-2005) simulated with JSBACH (see S1). Using a reference level
determined from the last ten years of the historical simulation allows us to keep the standing
wood on the present level and to account for the dependence of forest growing stocks (carbon
pools) to disturbances, silvicultural interventions and varying environmental conditions. Under
the GB harvesting rule, the wood harvest is only allowed to reduce the wood carbon pool down
to the reference level. Aside from environmentally driven decreases, the wood carbon pool thus
nearly remains constant over the whole simulation time.

2.4. Simulation runs with JSBACH
We conducted six simulations (Table 1) from 2006-2100, all starting from the same initial state
(see S1). The simulations differ in the applied harvest rule and in their climate and $CO_2$ forcing.
While the different RCP harvest maps were applied in simulations with the corresponding MPI-
ESM RCP forcing, each MPI-ESM RCP forcing was additionally run applying the GB
harvesting rule.
**Table 1**
2.5. Growth-based harvesting restricted to managed forests (MF)
To infer from the growth potentials simulated under GB how much biomass could potentially
be harvested (harvest potentials), we conduct a post-processing step overlaying a map that
masks out forest areas subject to conservation, infrastructural limits, or not being influenced by
human activities so far due to other reasons (Kraxner et al., 2017). Applying nearest neighbor
interpolation on the 1 km$^2$ spatially explicit map of primary forest intensity (0%-100%; Fig. 6
in Kraxner et al., 2017) we derived a T63 map of primary forest area. This static map was used
to filter the growth-based harvest determined in the GB simulations for 2006 to 2100, to only
account for managed forests (MF) in the mitigation assessment.
2.6. Analysis of wood harvest impacts on forest disturbances and natural mortality
To analyze the mechanisms driving differences in GB and RCP wood harvest amounts we can
formulate changes in above-ground wood carbon stocks over time (*dCw/dt*) as carbon gains
from net primary production allocated to the wood pools (*NPPw*) minus losses due to natural
disturbances and anthropogenic management (i.e., wood harvest, *h*):
$\frac{dCw}{dt} = NPPw - \frac{Cw}{\tau} - h$  (1)
In this conceptual formulation, the loss due to natural disturbances depends on the size of the
carbon stock and a time constant ($\tau$). As net primary production in our model does not depend
on harvest, GB growth potentials ($p_{GB}$) and RCP harvest can be related as
$p_{GB} = h_{RCP} + \left(\frac{Cw_{RCP}}{\tau_{RCP}} - \frac{Cw_{GB}}{\tau_{GB}}\right) + \left(\frac{dCw_{RCP}}{dt} - \frac{dCw_{GB}}{dt}\right)$  (2)
The amount of growth potential under GB can thus be split into several terms: The first term is
the reference harvest rate of the RCPs. The second term accounts for the difference in loss due
to natural disturbances in the RCP and the GB simulation. In JSBACH this can further be split
into differences in losses due to background mortality, such as self-thinning of forests, due to
fire, and due to windbreak. JSBACH explicitly integrates two modules for the simulation of
fire and wind disturbances depending on climate and carbon pools. The third term accounts for
the changes in the above-ground wood pool realized over time in the simulations. As shown
below, the RCP harvest results in an increase of above-ground woody biomass over the $21^{st}$
century for all three scenarios. For GB, on the other hand, $dCw_{GB}/dt$ should theoretically be
close to zero over time as GB aims to sustain the above-ground carbon pools of woody PFTs;
however, reductions in NPP due to less favorable climatic conditions or increased disturbances
can entail negative $dCw_{GB}/dt$. To summarize, GB includes the RCP wood harvest and,
moreover, makes use of additionally accumulated carbon and eventually reduced mortalities to
adapt harvest decisions to the novel climate and forest growing conditions.

225       2.7. Accounting for the mitigation potential of forest management in the Earth system

We account for long term effects of wood harvest, as in IAMs, by approaching a life cycle
analysis. Many wood products have lifetimes of decades to centuries. Here, we assess the effect
on atmospheric carbon content when harvested carbon is transferred, at least to a part, to longer-
lived product pools, instead of entering the atmosphere immediately. We compare this
"mitigation effect" achievable by the wood products harvested under the GB concept after the
map of managed forest area is overlaid (MF) to those achievable according to the three RCP
harvest maps. To this end, we distinguish three anthropogenic wood product pools -- bioenergy,
paper, and construction -- with 1, 10, and 100 year life times, respectively, as are typically
assumed in global modeling studies (Houghton et al., 1983; McGuire et al., 2001).
To allocate the wood biomass harvested in our JSBACH simulations to different product pools,
we made use of FAO country-specific statistics reporting wood production in fourteen different
categories (FAOSTAT, 2016). For our analysis, we assume that the production technology and
allocated percentage of each country's total wood production to these fourteen categories
remains constant at 2005 levels over the 21$^{st}$ century and used these percentages to allocate
wood biomass harvest from JSBACH (remapped to countries - see a calculus example in
supplementary material S2). The fourteen categories are then assigned to the three distinguished
anthropogenic wood product pools. We assume that the harvested material entering one of these
three product pools in a year decays at a rate of 1/lifetime, i.e. that all material used for
bioenergy is respired to the atmosphere within the same year it is harvested, while the material
entering the paper and construction pool is emitted at a constant rate over the following 10 or
100 years, respectively. The emissions at a given year for paper and construction pools are
therefore composed of a fraction of that year's harvest, but also of the legacy of material
harvested earlier, yielding annual emissions E from all three product pools as follows:
$$E(t) = f_b h(t) + \sum_{s=t-9}^{t} \frac{1}{10} f_p h(s) + \sum_{s=t-99}^{t} \frac{1}{100} f_c h(s) \qquad (1)$$
Here, f for bioenergy (b), paper (p), and construction wood (c) are the fractions with which the
harvested biomass is assigned to the product pools (see S2). We call E "emissions from product
decay" in the following.
To account for the fact that the emissions from product decay leave the atmosphere over time
to be taken up by the terrestrial biosphere and the ocean, we apply the response function (Eq.
2) by Maier-Reimer and Hasselmann (1987). Convolution of this response function with the
time series of annual emissions from product decay until year t results in the change in
atmospheric carbon content in that year, C(t) (Eq. 3).

$$G(t) = A_0 + \sum_{p=1}^{4} A_p \, exp^{-t/\tau_p} \qquad (2)$$
$C(t) = \int_0^t G(t-s) \cdot E(s)ds$                                                                     (3)

Emissions are present in the atmosphere as they occur and, therefore, $G(0) = 1$ and $A_0 = 1 -$
$\sum_p A_p$. The constants $A_p$ and the time constants $\tau_p$ are fitted for $p > 0$ using one of the best fits
found by Maier-Reimer und Hasselmann (1987): the sum of four exponential terms with time
constants $\tau_1$, $\tau_2$, $\tau_3$ and $\tau_4$ of approximately 1.9, 17.3, 73.6, and 362.9 years, and constants $a_1$,
$a_2$, $a_3$ and $a_4$ of 0.098, 0.249, 0.321, and 0.201. Accordingly, Equation 2 is an exponential
function that accounts for the uptake of $CO_2$ by ocean and land over time and Eq. (3) integrates
the accumulated amount of total $CO_2$ concentrations in the atmosphere at each time step
regarding past and present emissions. The mitigation effect of wood products is then determined
as the difference between the harvested material and the change in atmospheric carbon content.
3.   Results
3.1. Comparison of GB and RCP harvesting
Above-ground woody biomass is simulated to increase by the end of the 21[st] century for the
RCP wood harvest (Figure 1a), despite an increase of the amounts of wood harvest (Figure 1b).
This implies that the changes in environmental conditions lead to a larger accumulation of
woody biomass than is removed by the increased harvest. Depending on the RCP, the simulated
increase in above-ground woody biomass may reach 133% (425 PgC in 2100) of the initial
level in 2005 (320 PgC) for RCP8.5 and substantially higher levels of 128% and 117% for
RCP4.5 and RCP2.6, respectively (Figure 1a). The temporal pattern of this increase, with strong
increase only in the first half of the century for RCP2.6 or throughout the century for RCP8.5,
reflects the projected evolution of changes in $CO_2$ and climate (Collins et al., 2018).
For the GB rule, woody biomass remains more or less constant over time (Figure 1a), as the
average annual increment is removed by harvest by definition of the GB rule (see Methods).
Consequently, the growth potential of global forest resources under GB is simulated to be as
high as 9 PgCy$^{-1}$ at the end of the century subject to the realization of RCP8.5 climatic
conditions, or about 4 to 6 PgCy$^{-1}$ for the other two scenarios (Figure 1b). About two thirds of
the growth potential lie in managed forest areas and are thus potentially harvestable (Figure 1b,
MF-harvest curves). The MF harvest potentials are thus twice to three times (3-7 PgCy$^{-1}$) as
high as those of prescribed wood harvest simulated by IAMs for the RCPs. Note that, as
described in the methods, managed forest areas refer to the present-day state and may expand
in the future, which would further increase the harvest potential. These figures are harvestable
wood biomass amount and differ from commercially useable timber including bioenergy, paper,
and construction woody biomass (see 2.7 and 3.3).
We map the geographical distribution of RCP harvest as well as growth and harvest potential
under the GB harvesting rule applied to all global forest (GB) and managed forest areas (MF)
to recognize regional hotspots (Figure 2). Central Latin America including the accessible parts
of the Amazon forests, large parts of North America, the accessible parts of central Africa,
eastern Asia and Europe including Russia can be recognized under all climate scenarios as
hotspots for allocation of simulated harvest activities. The large harvest potentials of the supply-
based harvest in the tropics contrast with the patterns of the demand-based RCP harvest; in
particular in RCP2.6 and RCP4.5 much of the global harvest is provided from eastern North
America, central Europe and East Asia. A reasonable proportion of GB harvest amount in the
tropics is masked out in MF as inaccessible forest area; nevertheless the tropics contribute a
large harvest potential from wood supply side in both GB and MF.
**Figure 1**
**Figure 2**
3.2. Separation of the processes underlying the growth potentials under future climate

308        scenarios

The harvest potential under the GB harvesting rule in JSBACH exceeds RCPs wood harvest
defined by IAMs not only because of taking into account changes in growth rates caused by
changed environmental conditions, but also due to avoided mortality and disturbances (see
methods section). Figure 3 shows the separation of the growth potential underlying the GB
harvest into changes in standing wood as compared to RCP harvest, avoided background
mortality, natural fire, and wind disturbances, and the amount prescribed originally by RCPs.
The largest contribution to the growth potential under the GB harvesting rules exceeding the
RCP harvest is the lower background mortality, which is directly related to lower accumulation
of woody biomass (see Figure 1a). This lower accumulation also leads to the decreased carbon
losses from fire and wind disturbances. Depending on the climate scenario (RCPs) the simulated
reduction of mortality and disturbances add up to 2-5 $PgCy^{-1}$ at the end of the century. Under
the RCP harvest, woody biomass is simulated to mostly increase beyond what is required by
the increasing harvest rates (see Figure 1). Harvesting this "surplus", i.e. the increase of
standing biomass over time by applying RCP harvest rates and harvesting less biomass than the
annual increment provides, also contributes to the larger growth potentials under the GB
harvesting rule. The temporal evolution is different from that of avoided mortality and
disturbances, reflecting the projected changes in $CO_2$ and climate. Greater fluctuation of the
growth potential compared to the RCPs' annual wood harvest amounts is because of the direct
dependency of the forest's productivity on climate fluctuations.
**Figure 3**
3.3. Mitigation potential of GB versus RCP wood harvest
We show the mitigation potential of forest resources in the 21st century under growth-based
harvesting of global forest (GB) and managed forest (MF) areas versus the RCP wood harvest
prescribed from IAMs in Figure 4. Due to the larger harvested amounts, the mitigation potential
is higher for GB and MF compared to RCP harvest and the magnitude depends on the
underlying climate scenario. The advantage of growth-based harvesting lies in storing a larger
amount of carbon in wood products whilst keeping above-ground woody carbon pools constant.
These aspects are largely ignored by IAMs. Table 2 below shows the net mitigation potentials
of world forest resources (GB and MF against RCP harvest) by wood harvest at the middle and
end of the 21$^{st}$ century (2050 and 2100). The highest mitigation effect is achieved in the GB8.5
scenario with 140.6 PgC and 379.1 PgC up to 2050 and 2100, respectively. These figures
account for 278% and 287% more global carbon storage than in the RCP8.5 scenario with
prescribed RCP wood harvest with 50.6 and 132.1 PgC mitigation up to 2050 and 2100,
respectively. Only considering current managed forests, the mitigation effect realized for MF8.5
still reaches a maximum mitigation potential of 109.3 and 295.8 PgC up to 2050 and 2100,
respectively.
**Table 2**
**Figure 4**

347       4. Discussion

RCPs define wood harvest in each region according to scenarios realized by IAMs about social
and economic developments in the 21st century, but independent of ecological capacities of
forest ecosystems (van Vuuren et al., 2011; Riahi et al., 2011; Hurtt et al., 2011; Stehfest et al.,
2014). Although the growth-based harvesting rule realizes potentially a larger wood harvest
amount than the RCPs, it remains as per definition a sustained-yield forest harvesting approach
and guarantees sustainability of the current ecological conditions at each region with respect to
standing biomass. However, as a consequence, regions with low standing biomass, for example
due to extensive historical harvest, will maintain these low biomass levels. Below we discuss
the effectivity of GB in adapting to new environmental conditions and the mitigation potential

and highlight the missing issues in our simulation analysis, especially about the provisioning of multiple goods and services (e.g. biodiversity, forest health), and the future research themes about integration of diversified management strategies in ESMs.

Accounting for the climate state in simulating future forest harvest is crucial (Temperli et al., 2012; Sohngen and Tian, 2016). Accordingly, the novelty of the applied GB in this study is the dynamic nature of this management approach based on the ecology of forest ecosystems and climatic and atmospheric conditions. According to Schelhaas et al. (2010), an accelerated level of wood harvest to reduce the vulnerability of European old forests to wind and fire disturbances is needed to stop the current built-up of growing stock. Applying GB in this study realized an increased wood harvest rate for European forests (see Figure 2) showing first signs of carbon sink saturation and high vulnerability to natural disturbances (Nabuurs and Maseraet, 2013). Global studies of this nature are largely missing due to the lack of data and forest ecosystems complexity on global scale. Our idealized simulations suggest that GB does not only effectively safeguard sustainability of the current forest biomass on the global scale, but also positively affects the resistance of forest resources against natural disturbances and efficiently utilizes forest growth and productivity potentials (see Figure 3). Our estimates are, of course, sensitive to the choice of reference level: In this study, we applied the maximum current (1996-2005) above-ground wood biomass as the reference level. Any changes in this reference may affect the realized harvest potentials and should be carefully defined regarding ecological potentials and economic implications.

In our simulations, future environmental changes are mostly beneficial for accumulation of forest biomass, apparent from increasing standing biomass in the RCP harvest scenarios or the increase of GB and MF harvest rates over the 21$^{st}$ century. This is in line with other studies projecting above-ground forest carbon storage to increase in the future (e.g. Tian et al., 2016). These effects of environmental changes on forest growth are largely missing in the IAMs

providing the wood harvest scenarios to dynamic global vegetation models (DGVMs) and ESMs (see our description of IAMs in Sec. 2.2). IAMs do not account for the fact that the demand side may also be influenced by the availability of the resource and, accordingly, the increased biomass stocks projected for the future would likely lead to larger wood harvest rates than IAMs simulate by assuming present-day growth conditions. The extent to which accounting for environmental changes may influence estimates of harvestable material (e.g. apparent from comparisons of GB and MF harvest potentials under RCP2.6 as compared to RCP8.5, see Figure 1) highlights the need to include these effects in models, such as IAMs, that estimate future wood harvest. Our study is limited to considering biomass growth, albeit in interaction with soil conditions also responding to the altered climate. In reality, harvest decisions would consider further variables that depend on environmental conditions, such as the maximum soil expectation value, which are not explicitly simulated neither in our model nor in IAMs.

Note that the estimates of GB wood harvest as provided by our model are not meant as plausible estimates of actual future harvest, which as described before depends not just on resource availability and accessibility of areas, but is demand driven by other economic and political considerations. Limiting GB to available managed forest area, MF realized less harvest potential than GB, however, still a larger amount than RCP and with a higher mitigation potential (see Table 2). Also, actual future harvest will interact with other land-use decisions such as changes in forest cover due to agricultural expansion, but also afforestation. We have further not accounted for the effects of wood harvest on biodiversity, forest health, and other ecosystem services. Chaudhary et al. (2015) state that the effect of forest management on the species richness, for example, highly depends on the management regime applied. They refer to literature reporting a positive effect of logging activities on species richness as a result of establishing early successional colonizers. Additionally, applying selective logging approaches (e.g. future crop trees of targeted species) for forest management may enhance forest recovery

and reduce unintended changes in species composition (Luciana de Avila et al., 2017).  Instead
of actual forest harvest that considers all these aspects in its decision-making, our study
provides an estimate of the ecological potentials for wood harvest. However, the change in
resource potentials with climate change forms the ecological basis for realistic decision-making.
There is uncertainty in simulating ecosystem response to environmental changes. Regional
forest inventories show an increase in biomass due to historical environmental changes
(excluding effects of land-use change) (Pan et al., 2011). The largest sinks are found by these
studies to be in the tropical regions, coinciding with our simulated regions of largest potentials
for additional wood harvest. Also the other regions showing larger potential for wood harvest
under GB and MF than RCP, such as North America, Europe, Russia and East Asia, currently
exhibit carbon uptake due to historical environmental changes. This gives some confidence in
the robustness of our results, in particular since most models project the carbon sink in
vegetation to continue for the future; however, its magnitude is uncertain (Sitch et al., 2008). A
large source of uncertainty is the strength of the $CO_2$-fertilization effect (Kauwe et al., 2013;
Hickler et al., 2015;), which reflects in a large spread across models in estimates of global total
(vegetation plus soil) terrestrial carbon stocks (Arora et al., 2013) and of vegetation productivity
(Zaehle et al., 2014). To better assess these effects, we additionally simulated the future GB
and MF harvest potentials under present-day climate and $CO_2$ conditions (see simulations of
GBpd and MFpd in the supplementary material (S1)). These simulations led to a wood harvest
potential larger than that with RCPs harvest rates and rather constant harvest over time (~3.2
and 2.7 PgC annually for GBpd and MFpd, respectively, see S1-Figure 1). The harvest amount
of GBpd is equal to RCP8.5 harvest amount at the end of the century in our simulations.
Differences between GBpd and MFpd and the simulations forced by the different RCPs as well
as differences among the latter illustrate the effects of changes in climate and $CO_2$ concentration
on forest growth and resulting harvest potentials. The differences in wood harvest amounts
between the harvest simulations based on GB and MF and those with prescribed RCP wood
harvest rates in the first simulation year show differences of applying the supply-based harvest
rule (GB and MF) versus the demand-based RCPs under current environmental conditions. The
geographic allocation of growth and harvest potentials for GBpd and MFpd (see S1-Figure 2)
resembles those under RCPs, however, with higher global values. That the GBpd and MFpd
harvest potential are higher than the RCP harvest implies that the larger potentials as compared
to RCP harvest are partly attributable to the harvest simulated by IAMs not using the full
sustained, ecological potential (e.g. due to real-world demand). However, the harvest potentials
under RCP climate all grow substantially larger than the harvest potential under present-day
climate. This depicts the isolated effect of environmental changes, particularly $CO_2$ fertilization,
on the simulated potential harvest.

A further uncertainty in the model we used is that our model did not explicitly account for a
nitrogen cycle. Nitrogen may become a limiting factor for the additional uptake of carbon in
vegetation, although future climate change might also lead to higher nutrient availability due to
faster decomposition rates (Friedlingstein and Prentice, 2010). Further, nitrogen deposition may
reduce nitrogen limitation (Churkina et al., 2010), and it is not predictable if artificial
fertilization of managed forests may find wide-spread application in the future. Overall,
therefore, quantifications of effects of future climate change on global carbon stocks derived
from individual models have to be treated with care. Our model includes present-day nitrogen
limitation implicitly by choice of photosynthetic parameters and includes structural limits
prohibiting development of wood densities beyond observational values. Tests with a similar
model version as ours but representing an explicit nitrogen cycle suggest a rather small
sensitivity of the land carbon cycle to nitrogen limitation under $CO_2$ increases and climate
changes in the range of the RCP scenarios investigated here (Goll et al., 2017). The increase in
gross primary production (GPP) over the industrial era of our model (or similar versions) lie at
the high end, but within the range of a wide range of other models (Anav et al., 2013); recent
evidence from long-term atmospheric carbonyl sulfide (COS) records shows that models with
high GPP growth are most consistent with observations (Campbell et al., 2017). The location
of the largest potentials of GB and partly MF harvest simulated in our study being in the tropical
forests is consistent with the large carbon sinks derived from inventories for past environmental
change (Pan et al., 2011).
GB harvest was simulated to mitigate 255-380 PgC, depending on the realized RCP, through
wood product usage for the period 2006-2100 from global forest resources. Moreover, it
accounted for sustaining the above-ground wood carbon pool at the reference level of 1996-
2005. A comprehensive mitigation study, however, should take into account the total carbon
balance of forest ecosystems including soil plus litter carbon. Growth enhanced by
environmental changes, as simulated to lead to accumulation of woody biomass in the RCP
harvest simulations (Fig. 1a), may lead to larger input to the soil (if not removed by wood
harvest). However, soil carbon pools respond differently to environmental changes than forest
biomass. In particular, soil carbon models generally assume enhanced soil respiration under
higher temperatures (Friedlingstein et al., 2006), which may substantially offset the additional
carbon uptake by the vegetation (Ciais et al., 2013). As these processes act the same in our
simulations of GB, MF and RCP harvesting rules (as they share the same climate scenarios),
effects of environmental changes on soil carbon will likely not substantially affect our
comparison of GB, MF and RCP harvest in relative terms, but may alter the net carbon balance
in each of them. Further, the usage of wood products implies removal of carbon off-field. This
can lead to depletion of soil plus litter carbon stocks. Observational data generally found small
decreases of soil carbon, but substantial reduction of deadwood material (Erb et al., 2017). Such
effects must be expected to be stronger for GB and MF harvest with its larger harvested biomass
than for RCP harvest, reducing on-site carbon stocks, but consequently also soil respiration.
Estimating a mitigation potential based on the net carbon balance of vegetation, soil plus litter
and product pools therefore would depend on the actual size of soil and vegetation carbon pools

and the lifetimes of products relative to the lifetimes of the on-site carbon, which are further subject to a changing climate. There is not a unique life time for anthropogenic wood products pools in the literature. Lifetime of construction wood, for example, spanning from 67 years in Härtl et al. (2017), up to 160-200 years in van Kooten et al. (2007) are applied in recent studies. Regarding global variation of carbon turnover rate, Carvalhais et al. (2013) find mean turnover times of 15 and 255 years for carbon residing in vegetation and soil near to Equator and higher Latitude over 75°, respectively. Regarding the uncertainty about life time of anthropogenic wood pools, we stay consistent with the applied figures in FAO statistics (FAOSTAT, 2016) and other land carbon budget studies (Houghton et al., 1983; McGuire et al., 2001).

Despite carbon fluxes being the focus of land-use change as mitigation tool (e.g., UNFCC, 2012), forest management may enhance or mitigate climate change by a range of other mechanisms such as a change in surface albedo (e.g., Rautiainen et al., 2011; Otto et al., 2014) or turbulent heat fluxes (e.g., Miller et al., 2011). Such biogeophysical effects needed to be accounted for in a complete assessment of the mitigation potentials, as has been done for global land cover change (Pongratz et al., 2011) or for forest management on local (Bright et al., 2011) or regional scale (Naudts et al., 2016). These biogeophysical effects are particularly important for the local climate (Winckler et al., 2017). In our study, we restrict estimates of mitigation potential to carbon fluxes only and thus focus on the perspective of mitigating global greenhouse gas concentrations. This further allows for a direct comparison of the wood harvest scenarios provided as part of the RCPs.

Different from economic models, ESMs do not consider costs associated with early mitigation measures and thereby implicitly assume a zero social discount rate, meaning that there is no preference for immediate mitigation. However, the discount rate plays a major role to find economically the most efficient mitigation action (Stern, 2007). van Kooten et al. (1999) analyzed the sensitivity of investments for carbon sequestration to discount rate in western Canada and found that applying zero discount may not provide enough incentive for increasing

carbon storage. However, most forest carbon cost studies are inconsistent in using terms, geographic scope, assumptions, program definitions, and methods (Richards and Stokes, 2004) and may not truly assess carbon sequestration potentials of forest ecosystems. Therefore, if there were a social preference for prompt climate change mitigation, carbon sinks later in the century should be discounted. Regarding the discussion on discount rate, Johnston and van Kooten (2015) argue that applying sufficiently high discount rates in substituting biomass for fossil fuels never leads to carbon neutrality.

## 5. Conclusions

We recommend that future research on integration of management strategies in DGVMs and ESMs should regard ecological sustainability as well as socio-economic challenges. In reality and today, forest management is more of a gamble than a scientific debate (Bellassen and Luyssaert, 2014) and there is no consensus in applying a certain forest harvest rule (e.g. GB) among forest owners, decision-makers and local users. The rationale to manage forest resources sustainably and efficiently is generally recognized and implemented (Luckert and Williamson, 2005; Elbakidze et al., 2013). However, the process of forest management decision-making is based on the past experiences with a business-as-usual strategy (BAU). Adaptation to future environmental change and minimizing the risks associated with climate change impacts is recently fully integrated in forest research (Lindner et al., 2014), however, remains in experimental level in implementation (Yousefpour and Hanewinkel, 2015). Mitigation, in turn, is of public interest and there are some attempts internationally to account for mitigation effects of forest management in carbon policy. International programs such as the Kyoto protocol encourage forest managers to store carbon in the forest stocks on the ground applying financial instruments such as tax reduction and direct purchase of carbon offsets. Therefore, inclusion of financial aspects in global forest management modelling and decision-making may help to put scientific results into practice (Hanewinkel et al., 2013). This suggestion is in line with van

Vuuren et al. (2011) about the necessity of strengthening the cooperation between integrated assessment models (IAMs) and Earth system modelling communities to improve the understanding of interactions and joint developmet of environmental and human systems. Our study is the first implementation to account for the climate-dependence of forest growth on global scale for harvest potentials. It suggests the importance of considering this dependence: the growth-based harvest approach (GB) as applied in this study may realize wood harvest potentials twice to four times as high as those of prescribed wood harvest simulated by IAMs for the RCPs and would closely triple the net mitigation effects of wood products. By limiting GB to managed forests (MF), we simulated a lower harvest potential than GB, still two to three times more than in the IAMs, which could double the net mitigation effect of wood harvest potential in the 21$^{st}$ century. To move from estimates of potentials to actual harvest rates, climate-dependent forest growth needs to be integrated with socio-economic factors to fully incorporate economic aspects of forestry practices within a dynamic forest growth and yield modelling system.

Code availability

Scripts used in the analysis and other supplementary information that may be useful in reproducing the authors' work are archived by the Max Planck Institute for Meteorology and can be obtained by contacting publications@mpimet.mpg.de.

Data availability

Primary data are archived by the Max Planck Institute for Meteorology and can be obtained by contacting publications@mpimet.mpg.de.

Sample availability

None

Appendices

None

Supplement link (will be included by Copernicus)

Supplementary includes two main files: a word document S1 on the "details on JSBACH, the model version, the simulation setup, and the additional simulation with present day forcing" and a zipped excel file S2 as an example of how "mitigation potentials of woody products in their life time" are calculated.

Team list

See authors list

Author contribution

R.Y. and J.P. initiated the study. R.Y. performed the model simulations. J.N. implemented the code changes. All authors contributed to analyzing the simulations and writing the manuscript.

Competing interests

The authors declare that they have no conflict of interest.

Acknowledgments

This work was supported by the German Research Foundation's Emmy Noether Program (PO 1751/1-1). Computational resources were made available by the German Climate Computing Center (DKRZ) through support from the German Federal Ministry of Education and Research (BMBF).

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

Figure's Captions
**Figure 1 Development of global standing wood carbon pools forced by three different RCP**
**scenarios and subject to the harvesting rules of the representative concentration pathways**
**(RCP2.6, RCP4.5 and RCP8.5) or subject to growth-based harvesting (GB2.6, GB4.5, and**
**GB8.5) (1a). Development of RCP wood harvest rates, of the growth potential of forests**
**under GB, and of the harvest potential under GB limited to global managed forest area**
**(MF2.6, MF4.5, and MF8.5) (1b). All lines are smoothed over 10 years.**
**Figure 2 Spatial distribution of the harvest realized in JSBACH when harvest rates are**
**prescribed from the representative concentration pathways (left panels), of the harvest**
**potential applying the growth-based harvesting rule to available managed forest area**
**(right panels) and of the underlying growth potential (middle panels). All values are**
**summed over the entire simulated period (2006-2100).**
**Figure 3 Composition of growth-based harvest (GB) forced by different climate change**
**scenarios (RCP2.6, RCP4.5, and RCP8.5 in figures a, b, and c, respectively).** *dCw/dt* **refers**
**to the difference in changes in above-ground woody biomass between representative**
**concentration pathways' and GB harvest (where changes in biomass in GB are by**
**construction of the harvest rule close to 0), BGmort refers to the difference in woody**
**carbon losses between RCP and GB harvest due to background mortality, Fire to that due**
**to fire disturbance, and Wind to that due to wind disturbance. GB and RCP harvest are**
**as in Figure 1b.**
**Figure 4 Net mitigation potentials from the growth potential under the growth-based**
**harvesting rule (GB) (a, b, c), representative concentration pathways' (RCP) harvest (d,**
**e, f), and GB harvest limited to managed forest area (MF) (g, h, i). Left axes show the**
**annual carbon fluxes due to harvested material and product decay changing atmospheric**
**CO$_2$ concentration, and the mitigation potential of wood products as the difference of**
**both. Right axes accumulate the annual figures over time.**

**Table 1: JSBACH simulations conducted in this study with the applied harvesting rule and climate and $CO_2$ forcing.**

| Name | Harvest rule | MPI-ESM forcing |
|------|-------------|-----------------|
| GB2.6 | GB | RCP2.6 |
| GB4.5 | GB | RCP4.5 |
| GB8.5 | GB | RCP8.5 |
| RCP2.6 | RCP2.6 map | RCP2.6 |
| RCP4.5 | RCP4.5 map | RCP4.5 |
| RCP8.5 | RCP8.5 map | RCP8.5 |

**Table 2 Net mitigation potentials of GB, MF and RCP harvest at the middle and end of**
**the 21st century**

| Applied harvest rule | Harvested wood (PgC) | | Mitigation effect (PgC) | |
|---|---|---|---|---|
| | 2050 | 2100 | 2050 | 2100 |
| RCP2.6 | 58.1 | 137.6 | 38.3 | 85.1 |
| RCP4.5 | 62.9 | 147.2 | 40.7 | 90.2 |
| RCP8.5 | 76.5 | 211.8 | 50.6 | 132.1 |
| GB2.6 | 192.7 | 421.3 | 124.5 | 255.0 |
| GB4.5 | 210.0 | 513.9 | 136.4 | 314.7 |
| GB8.5 | 215.0 | 609.4 | 140.6 | 379.1 |
| MF2.6 | 148.3 | 324.3 | 96.6 | 199.5 |
| MF4.5 | 161.6 | 395.1 | 105.6 | 244.9 |
| MF8.5 | 166.4 | 472.9 | 109.3 | 295.8 |



**Figure 1**

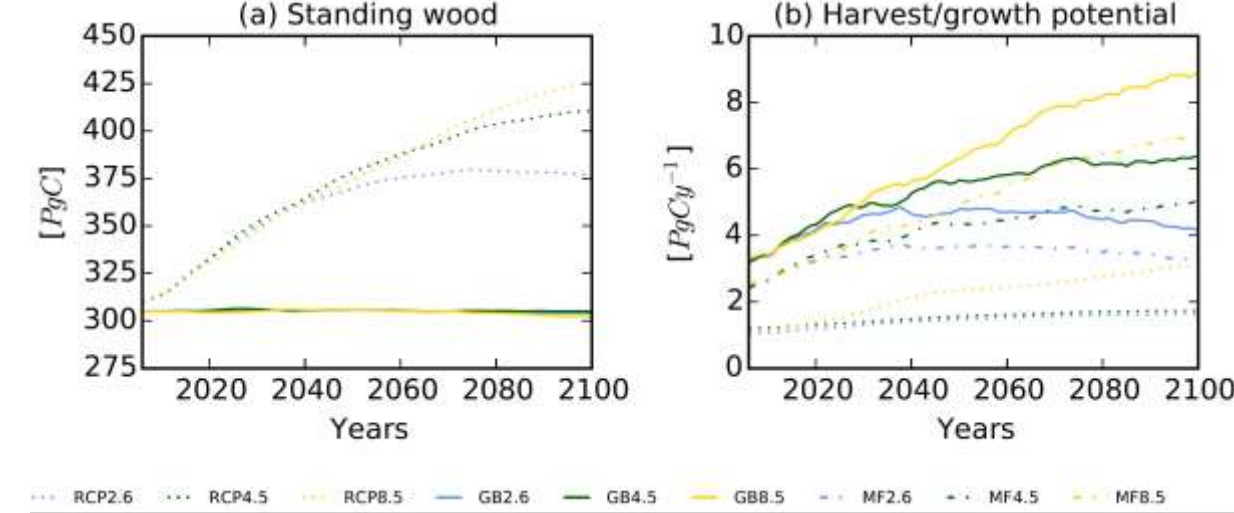



**Figure 2**

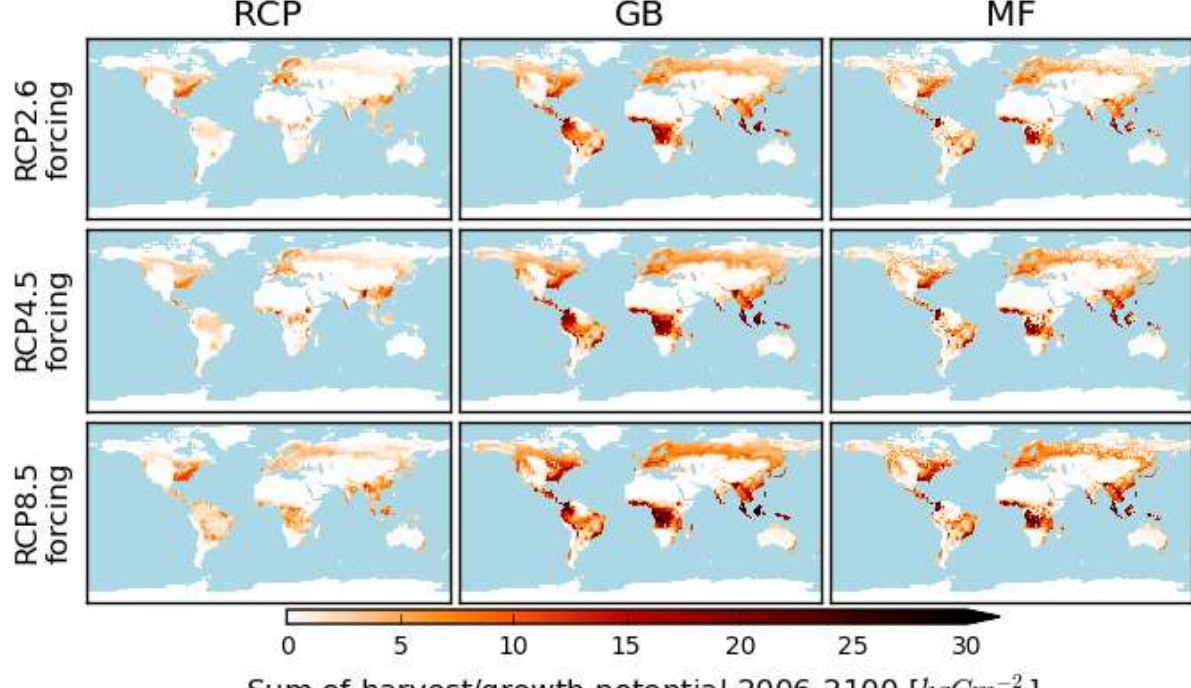

Sum of harvest/growth potential 2006-2100 $[kgCm^{-2}]$



**Figure 3**

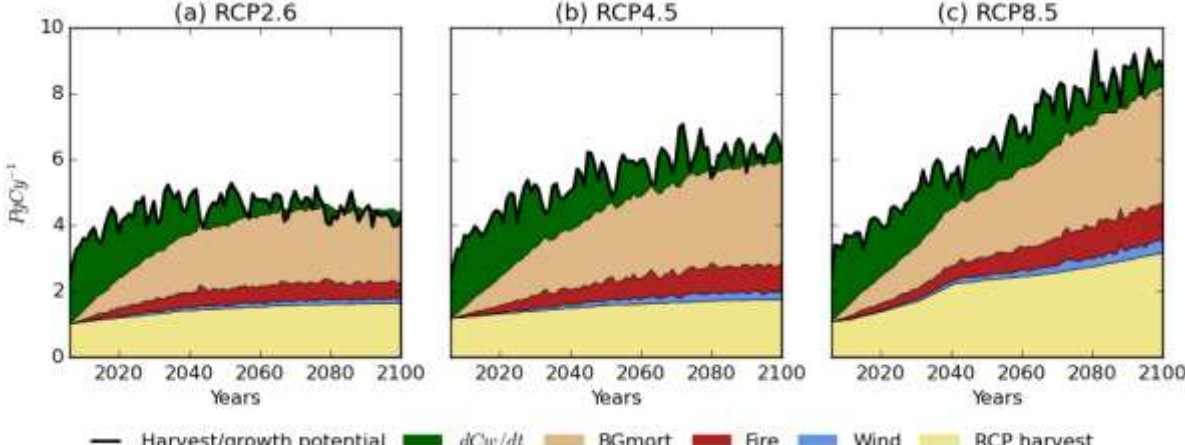



**Figure 4**

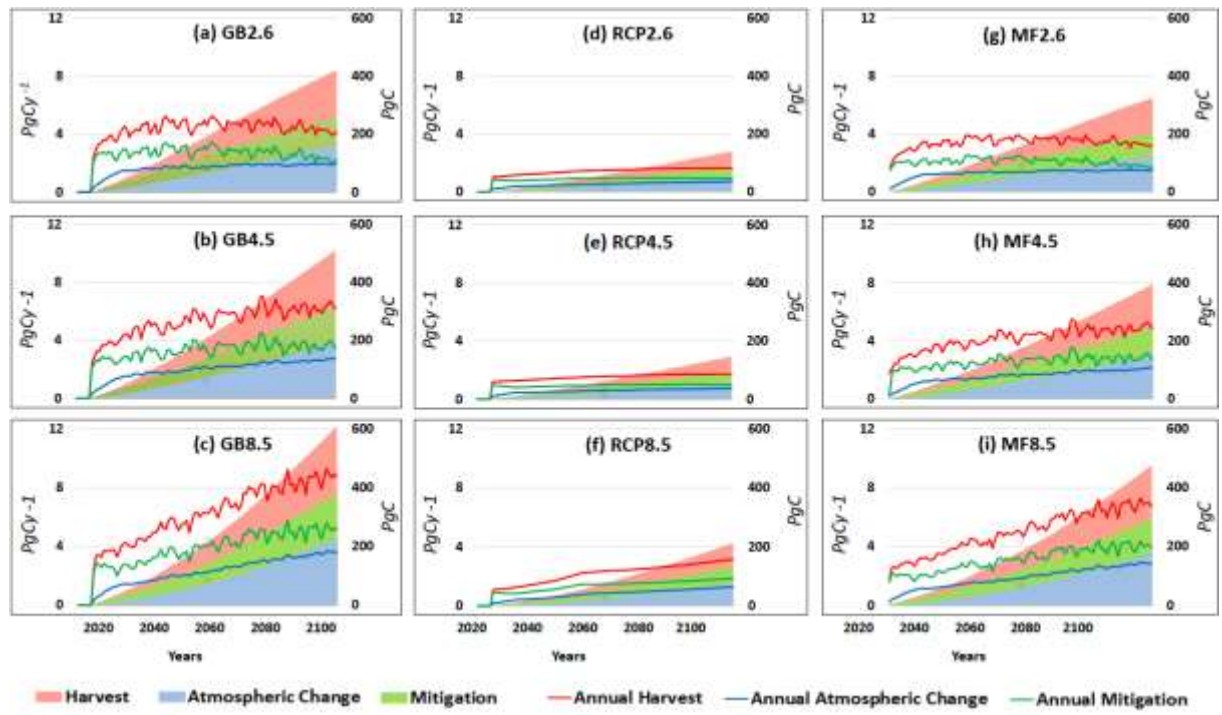

