# Peer review of "Title Page 1 2 Title: Simulating growth-based harvest adaptive to future climate change 3 List of authors: Rasoul Yousefpour1,2\*, Julia E.M.S. Nabel1, Julia Pongratz1,3 4 1 Land in the Earth System, Max Planck Institute for Meteorology, Bundesstr. 53,"

_Biogeosciences, 2017_

## Referee Comment (RC1) · Anonymous Referee #1 · 15 Feb 2018

The authors use a state-of-the-art Earth System Model (ESM) "JSBACH" to show that a revised harvest schedule in order to keep the forest carbon pools constant under climatic change will lead to harvest amounts twice to four times larger than those scheduled under standard harvesting rules, with the latter being based on Integrated Assessment Models. The net mitigation effects over the 21st century would then be much higher than under standard assumptions.

This paper is well written and illustrated and the results stress the importance of forest management, but there exists concern about the assumptions made to generate these results. The assumptions appear in part too artificial, so that the conclusions made are possibly not all well justified.

General comments:
1) Harvesting always the timber growth (increment) is a famous traditional forest management rule, which foresters have tried to apply already for more than one century, but which is hardly possible to be achieved. For example, following an empirically/iteratively based form of forest planning, the so called "Kontrollmethode" has been developed for forest management already at the end of the 19th century by de Liocourt (France) and Biolley (Switzerland). Implementation of such traditional harvesting rule into a state-of-the-art ESM has some originality, although it appears rather unrealistic that the assumption to apply this rule may ever be a realistic guide for real world forest management. This would also raise doubts about its usefulness for aggregated global projections.

In forest management the timber increment may only be measured post hoc and past increments can hardly be used to predict future timber increments. The historical and future timber increment depend both much on the actual forest structure, which is continuously changing. But this is only one limitation for the application of the modelled harvesting rule. Even more important is the in this study disregarded accessibility of the standing timber volumes in the world's forets. It may be totally uneconomic to harvest on steep mountain slopes or to carry out sustainable harvest (without deforestation) in hardly accessible tropical regions. Disregarding the probability with which a harvest will actually occur at a certain place will always lead to a great overestimation of the actually possible harvest. Such overestimation appears to be the case with the current paper as well. In conclusion, a better justification of the adopted harvesting rule and a more critical discussion of the results would be desirable.

2) One could consider it also as a limitation that the area occupied by a specific plant functional type have been kept constant, which means that an important element of dynamic vegetation modelling has been excluded. The assumption of a constant forest area over the next 100 years alone is very strong, meaning that land-use/cover change is ignoredover such long period. Furthermore, one of the most important tasks of forest management is to plan meaningful change of the forest composition, for example to
adapt to climate change. As mentioned already the structure of the harvests (size of harvested timber, tree species) would be important for the structure of the remaining timber volume. This would also have an impact on the timber growth. It is still a bit unclear how these complexities have been addressed. Alternatives to the harvesting algorithms applied in this paper should be discussed, or even better applied.

3) In addition, the notion that socio-economic rules to decide when and how much timber to harvest would always disregard actual environmental conditions is not fully valid. Often, the achieved timber size (Europe) or an economic criterion (international perspective), such as the maximum soil expectation value, are criteria to decide when to harvest. These criteria depend on the environmental conditions and could alternatively have been used to carry out predictions on timber harvests or to provide scenarios on a more realistic basis.

4) A central outcome of the submitted study is a much increased timber harvest particularly in the tropical forest biome (Figure 2). This result could be interpreted with more care. The tropical biome still comprises vast area of more or less natural forest, where NPP may actually be high. But NPP is not equal to commercial timber harvest and harvesting up to 25 - 30 kg C per square meter (2006-2100) in these forests would certainly destroy these ecosystems, with their particularly rich biodiversity. A harvest of 25 kg C per square meter would mean harvesting in the order of 1000 cubic meters per hectare over 94 years (or around 10 cubic meters per ha per year). This rough estimation implies 50% C content in dry woody biomass and an average timber density of 500 kg per cubic meter. Such harvest would be detrimental, as these natural ecosystems can often provide not much more than 0.5 cubic meter commercial harvest per hectare per year on a sustainable basis.

5) It appears that sustainable yield harvesting mainly reflects the NPP of the forests considered, because the size of the NPP appears to be harvested. The distribution of this NPP (see Figure 2) could be an interesting issue as well for a revised paper.
Should the harvesting aspect still be the main focus of an improved manuscript, one could consider the following recommendations:

More detailed issues:

1) The assumed lifetimes for wood products appear very high. They should be better justified and compared with those assumed for other studies, for example by Härtl et al. (2017) in Mitigation and Adaptation Strategies for Global Change. One could mention that there is still very large uncertainty concerning these values.

2) The sustainable yield scenario should be critically discussed considering the issues mentioned above. Moreover, some constraints could be considered. For example could the harvest in protected areas and in inaccessible forest areas be significantly reduced or even set to zero. For the purpose of a better prediction of the possible harvest, one could refer to the recent works by Luciana de Avila, e.g. "Recruitment, growth and recovery of commercial tree species over 30 years following logging and thinning in a tropical rain forest", which recently appeared in Forest Ecology and Management.

3) The discussion of discounting C over time is unclear. Also, more recent references could be included, such as Johnston and van Kooten (2015) "Back to the past: Burning wood to save the globe" published in Ecological Economics.

BGD

---

## Referee Comment (RC2) · Anonymous Referee #2 · 23 Feb 2018

Forests are one of Earth's most important resources and wood harvest is one of the main ways in which humans are managing and impacting forest ecosystems. This paper examines the question of how an alternative future wood harvest scenario, based on a "sustained yield" (SY) approach, and responsive to a changing environment, would differ from the standard demand-based wood harvest scenarios currently used by Integrated Assessment Models (IAMs).

Under the sustained yield approach, annual wood harvest rates are "optimized" and are equal to the annual rate of re-growth, thus keeping forest stocks maintained at their current levels. In addition, unlike most IAM wood harvest scenarios, the SY approach in this paper uses regrowth rates that are responsive to changing climate and $CO_2$ concentrations, thus potentially allowing higher wood harvest rates than the IAM wood

harvest scenarios that are based on static environmental conditions.

This work shows one way in which IAMs and ESMs could improve and strengthen their interactions. The paper describes the results of a set of simulations using the dynamic vegetation model JSBACH, forced with climate data from Earth System Model runs from three different Representative Concentration Pathways (and thus three different climate futures). The paper compares results from these JSBACH simulations using both the SY wood harvest scenario as well as the prescribed wood harvest scenarios from the IAMs.

This is an interesting thought experiment, and yet it is not really presented as such in this paper. In fact, one of my main criticisms with the paper is the way in which the work is framed and motivated. The SY approach is unrealistic and it is not clear why, or how, a society would want to pursue a wood harvest plan that involved harvesting more wood than demanded and doing so by harvesting every patch of forest by the exact amount it would regrow each year. It is only in the Discussion that the authors acknowledge that the SY scenario is not meant as a plausible estimate of future wood harvest, but rather as an estimate for the ecological potential for wood harvest. There is also not a lot of explanation given for why IAM demand-based wood harvest scenarios are inherently problematic (aside from not responding to changing environmental conditions). The motivation for using this particular SY approach appears to be a mix of exploring how changing environmental conditions could alter the amount of wood harvest, and also how additional wood harvest could act as a carbon mitigation method. However, the effect of changing environmental conditions on the amount of wood harvested is mixed with the effect of the additional wood harvest that is imposed by the SY approach (which attempts to harvest much more than the demand-based scenarios). In addition, the mitigation potential of the SY approach seems like a fairly inefficient way to capture and store carbon from biomass. The authors also do not address negative impacts of the SY scenario such as the impact on biodiversity (even partial removal of forest would have a negative impact on habitat), or the impact on the overall forest health that

could result from continued and widespread human management.

Due to the SY approach involving changing forest biomass in response to changing environmental conditions, as well as additional wood harvest to meet the forest regrowth rates, it is difficult to tell how much of the increased wood harvest is coming from the additional forest growth due to changing climate and $CO_2$ concentrations, and how much is coming from choosing to harvest more wood, and in more locations, than the IAM wood harvest scenarios. It would actually be quite interesting to look at this some more and I think the paper would benefit from an additional experiment that was devised to do this (as outlined below).

I think the paper would be much improved if a couple of key changes were made:

1) A better framing and motivation for the paper in the Introduction, to make it clear that the SY approach is a thought experiment to examine how much sustainable future wood harvest is possible, and why demand-based wood harvest scenarios are not sufficient.

2) In addition to the standard demand-based scenario, and the SY scenario, include a third wood harvest scenario that uses the prescribed demand-based wood harvest scenarios, but allows for forest regrowth rates to change due to changing environmental conditions, and thus allows for changes in the actual wood removed from forests. For example, this scenario could use the prescribed wood harvest area in each gridcell (instead of the wood harvest biomass), or it could use the ratio of prescribed wood harvest biomass to prescribed available forest biomass in each grid-cell. Either of these alternative wood harvest scenarios would retain most of the information from the prescribed demand-based scenario, but would allow the actual biomass harvested to change with changing environmental conditions. This could enable a simple quantification of the impacts of changing climate and $CO_2$ concentrations on future wood harvest, and would separate that effect from the effect of harvesting much more wood under the SY scenario.

---

## Author Comment (AC1) · 17 May 2018

The authors use a state-of-the-art Earth System Model (ESM) "JSBACH" to show that a revised harvest schedule in order to keep the forest carbon pools constant under climatic change will lead to harvest amounts twice to four times larger than those scheduled under standard harvesting rules, with the latter being based on Integrated Assessment Models. The net mitigation effects over the 21st century would then be much higher than under standard assumptions. This paper is well written and illustrated and the results stress the importance of forest management, but there exists concern about the assumptions made to generate these results. The assumptions appear in part too artificial, so that the conclusions made are possibly not all well justified.

[Figure]

Response: We appreciate the comment of the reviewer #1 agreeing on the importance of simulating forest management effects in a global study. Below we respond to the concerns regarding assumptions behind our experiment and all general and specific comments raised by the reviewer.

General comments: 1) Harvesting always the timber growth (increment) is a famous traditional forest management rule, which foresters have tried to apply already for more than one century, but which is hardly possible to be achieved. For example, following an empirically/iteratively based form of forest planning, the so called "Kontrollmethode" has been developed for forest management already at the end of the 19th century by de Liocourt (France) and Biolley (Switzerland). Implementation of such traditional harvesting rule into a state-of-the-art ESM has some originality, although it appears rather unrealistic that the assumption to apply this rule may ever be a realistic guide for real world forest management. This would also raise doubts about its usefulness for aggregated global projections. In forest management the timber increment may only be measured post hoc and past increments can hardly be used to predict future timber increments. The historical and future timber increment depend both much on the actual forest structure, which is continuously changing. But this is only one limitation for the application of the modelled harvesting rule. Even more important is the in this study disregarded accessibility of the standing timber volumes in the world's forets. It may be totally uneconomic to harvest on steep mountain slopes or to carry out sustainable harvest (without deforestation) in hardly accessible tropical regions. Disregarding the probability with which a harvest will actually occur at a certain place will always lead to a great overestimation of the actually possible harvest. Such overestimation appears to be the case with the current paper as well. In conclusion, a better justification of the adopted harvesting rule and a more critical discussion of the results would be desirable.

Regarding the comment 1): We agree that SY may serve as an experiment to simulate the ecological potential of global forest resources for producing wood and mitigating $CO_2$. We make it clear in the revised manuscript that the aim of our study is to estimate potentials, not actually possible harvest, and to show by the large changes in potentials under climate change that it is essential for models such as IAMs to capture the effects of climate change on harvestable material. We acknowledge that SY decisions are solely ecologically driven and economic factors such as wood prices are not considered. Global DGVMs mainly have a coarse resolution and application of spatially explicit forest harvest decision rules such as Control Method of Biolley is very limited if not impossible. We agree that it is worthwhile for future model development to regard this recommendation, as it is e.g. done in European studies (e.g. Naudts et al., 2016), if the aim is to estimate actual harvest. We make now clearer to the general reader that this study simulates the potentials of global forest resources for wood production, rather than actually possible rates. The changes in the manuscript are stating explicitly that our estimates refer to potential wood harvest rate throughout the manuscript, that our simulations should therefore be seen as thought experiments (see also comments by reviewer #2), and that, correspondingly, our assessment of mitigation potentials serves as a link of these potentials to $CO_2$ emissions and concentrations. Limiting ourselves to simulating ecological potentials of global forests in producing wood also means that we do not mask out non-accessible forests and protected areas. Therefore, and according to the suggestion by the reviewer #1, we bring this important issue not only at two occasions in the discussion but also outline it explicitly in the introduction to avoid misunderstanding.

Concerning the comment that past increments cannot serve to predict future increments: Indeed, our simulations capture that forest structure is changing over time. Only the target biomass stocks are derived from a fixed state: The present amounts of above-ground wood biomass of global forest resources serve as the harvest baseline for our SY decision rule. We make decisions based on the present stocks (over 10 years), and consider the future changes in growth (including regrowth after harvesting) of forests in deciding about the amount of wood harvest.

2) One could consider it also as a limitation that the area occupied by a specific plant functional type have been kept constant, which means that an important element of dynamic vegetation modelling has been excluded. The assumption of a constant forest area over the next 100 years alone is very strong, meaning that land-use/cover change is ignoredover such long period. Furthermore, one of the most important tasks of forest management is to plan meaningful change of the forest composition, for example to adapt to climate change. As mentioned already the structure of the harvests (size of harvested timber, tree species) would be important for the structure of the remaining timber volume. This would also have an impact on the timber growth. It is still a bit unclear how these complexities have been addressed. Alternatives to the harvesting algorithms applied in this paper should be discussed, or even better applied.

Regarding the comment 2): We agree that land use and land cover change, species composition and other adaptive measures in the future may change forest productivity and consequently the actually available material for wood harvest. Therefore, we discuss this important issue now in the paper. As stated in the manuscript, the simulations were conducted without dynamic vegetation and without land-use transitions to prevent changes in the areas occupied by the different PFTs and to be thus able to isolate the effects of forest management activities.

3) In addition, the notion that socio-economic rules to decide when and how much timber to harvest would always disregard actual environmental conditions is not fully valid. Often, the achieved timber size (Europe) or an economic criterion (international perspective), such as the maximum soil expectation value, are criteria to decide when to harvest. These criteria depend on the environmental conditions and could alternatively have been used to carry out predictions on timber harvests or to provide scenarios on a more realistic basis.

Regarding the comment 3): It is true that in reality such socio-economic factors are included in making a harvest decision. However, the IAMs used to project harvest rates in the representative concentration pathways do not allow variables such as maximum soil expectation value to change in response to altered environmental conditions. We add this discussion point to the manuscript.

4) A central outcome of the submitted study is a much increased timber harvest particularly in the tropical forest biome (Figure 2). This result could be interpreted with more care. The tropical biome still comprises vast area of more or less natural forest, where NPP may actually be high. But NPP is not equal to commercial timber harvest and harvesting up to 25 - 30 kg C per square meter (2006-2100) in these forests would certainly destroy these ecosystems, with their particularly rich biodiversity. A harvest of 25 kg C per square meter would mean harvesting in the order of 1000 cubic meters per hectare over 94 years (or around 10 cubic meters per ha per year). This rough estimation implies 50% C content in dry woody biomass and an average timber density of 500 kg per cubic meter. Such harvest would be detrimental, as these natural ecosystems can often provide not much more than 0.5 cubic meter commercial harvest per hectare per year on a sustainable basis.

Regarding the comment 4): It is of course true that only a fraction of the standing biomass in tropical forests is suitable for commercial timber harvest. It needs to be noted, however, that the harvest we simulate beyond that prescribed by the RCPs stems from the increase in standing biomass due to environmental changes, not from increasing harvest from the "baseline" biomass (baseline being present-day levels of harvest). This means that we do not reduce the current forest biomass, but harvest only biomass in excess of this, therefore not causing a degradation of standing biomass stocks. It is possible of course that it is mostly unusable plant species (such as lianas) that are responsible for the increase in biomass; however we have no reason to speculate that this is the case. It is possible that our model in general overestimates tropical biomass stocks, but this seems unlikely given evaluation studies revealing rather low vegetation carbon estimates for related model versions (Anav et al., J. Clim., 2013). It also needs to be noted that our harvest rates do not refer to commercial timber harvest, but also to fuel wood harvest, which can be fulfilled from a much wider range of biomass than commercial timber. As stated above, we have added statements to the manuscript (discussion part) that our estimates of harvest potentials does not consider biodiversity or conservation aspects and acknowledge that such considerations might lead to lower actual than potential harvest rates.

5) It appears that sustainable yield harvesting mainly reflects the NPP of the forests considered, because the size of the NPP appears to be harvested. The distribution of this NPP (see Figure 2) could be an interesting issue as well for a revised paper. Should the harvesting aspect still be the main focus of an improved manuscript, one could consider the following recommendations:

Regarding the comment 5): It is correct that the harvested wood increment is strongly related to the NPP of the considered forests and harvesting it is still the focus of the revised manuscript. We considered all recommendations given by the reviewer and state our replies to these detailed issues below.

More detailed issues 1) The assumed lifetimes for wood products appear very high. They should be better justified and compared with those assumed for other studies, for example by Härtl et al. (2017) in Mitigation and Adaptation Strategies for Global Change. One could mention that there is still very large uncertainty concerning these values.

Response: We justify the application of different life times to the anthropogenic wood pools and refer to the suggested paper and others to justify our application.

2) The sustainable yield scenario should be critically discussed considering the issues mentioned above. Moreover, some constraints could be considered. For example could the harvest in protected areas and in inaccessible forest areas be significantly reduced or even set to zero. For the purpose of a better prediction of the possible harvest, one could refer to the recent works by Luciana de Avila, e.g. "Recruitment, growth and recovery of commercial tree species over 30 years following logging and thinning in a tropical rain forest", which recently appeared in Forest Ecology and Management.

More detailed issue 2): We have extended the discussion section according to the recommendations by the reviewer, i.e. discuss in more detail SY effects on biodiversity, life cycle analysis, and carbon discounting. We have also clarified throughout the manuscript that our study aims at simulating potential rather than actual wood harvest rates (see also response to comment 1).

3) The discussion of discounting C over time is unclear. Also, more recent references could be included, such as Johnston and van Kooten (2015) "Back to the past: Burning wood to save the globe" published in Ecological Economics.

More detailed issue 3): We now discuss the social discounting of carbon using Johnston and van Kooten (2015) as reference.

Please also note the supplement to this comment:
https://www.biogeosciences-discuss.net/bg-2017-531/bg-2017-531-AC1-supplement.pdf

**Supplement:**

[revised manuscript text omitted]

S1: Supplementary material 1

A1. JSBACH simulations

Simulations were conducted with revision 7277 of cosmos-landveg-fom, a svn branch of revision 7215 of cosmos-landveg, the JSBACH development branch of the department "The Land in the Earth System" of the Max Planck Institute for Meteorology.

Simulations were executed on the IBM Power 6 machine BLIZZARD at the German Climate Computing Center (DKRZ).

Sustained-yield (SY) forest harvesting is implemented in this model version as described in the methods section of the main text. A modification over earlier JSBACH versions is that wood harvest applies just to harvesting from all woody PFTs and specifically from the above-ground wood carbon pool. To isolate the effects of different wood harvest rules, we do not apply land-cover change and dynamic biogeographic vegetation shifts for our future scenarios. We take into account changes in wood carbon pool, natural mortality and forest disturbances to determine the net annual increment of the above-ground wood carbon pool as the maximum amount to be harvested from forest areas.

A1.1 Initial state in 2006

All simulations described in the paper started in 2006 from the same initial conditions. These conditions base on a spin-up of the terrestrial system state using the MPI-ESM climate from the historical (1850-2005) CMIP5 experiment (Giorgetta et al., 2013) and land-use change and wood harvest data from Hurtt et al. (2011).

The initial state was derived carrying out three consecutive simulations. (I) An initial simulation with JSBACH to spin-up photosynthesis, phenology, hydrology and running climatic means required by the disturbance module of JSBACH. This simulation was forced by cycling the first years (1850-1879) of the historical CMIP5 experiment for four times. Wood harvest was fixed to the level of the initial year 1850 and no land-use change was applied. (II) A simulation with the stand-alone carbon cycle module of JSBACH to equilibrate the carbon pools with respect to the driving climate. This simulation was forced by NPP, LAI and climatic means, resulting from the preceding JSBACH simulation. (III) A second JSBACH simulation resuming the first JSBACH simulation, but starting from the equilibrated carbon pools. In this second simulation the full transient (1850-2005) climate from the historical CMIP5 experiment was used and land-use change and wood harvest were prescribed according to the data from Hurtt et al. (2011).

A1.2 Reference level for SY

An important decision for our study is the definition of the reference level of the wood carbon pool to be kept constant in the future applying SY. As one of the main goals of our study is to estimate potentials for wood harvesting under future climate scenarios, consistent with the historic past, we refer to the current level of wood carbon pools. The reference level for the SY simulations was therefore derived from the maximum simulated wood carbon per grid-cell and PFT in the period from 1996 to 2005 under the historical JSBACH simulation (see A1.1 simulation III). Because the historical JSBACH simulation was subject to land-use and land-cover changes maximum wood carbon densities were used instead of wood carbon stocks.

A1.3 Simulation of SY under present-day climate

This simulation (SYpd) keeps the current level of wood carbon pools constant as described above in A1.2. However, it is not forced by a transient but a cycled detrended present-day climate of the period 1991-2020 with a constant CO2 concentration (381 ppm) as the average value of the period (1991-2020). S1-Figure 1 shows the development of wood carbon pool and harvested amount resulting in the simulation SYpd compared to the 6 simulations described in the main text. SYpd realizes a higher wood harvest (+3.2 PgC) than RCPs (~1.2 PgC) at the beginning of simulations and equals the RCP8.5 wood harvest at the end of century. SYpd diverges from wood harvest amount by SY2.6, SY4.5, and SY8.5 largely towards the end of the century and remains below these figures. The geographical allocation of realized wood harvest amount as shown below in SYpd in S1-Figure 2 resembles largely the other SYs (see Figure 2 in the manuscript), however, the amount of harvested wood is lower.

**S1-Figure 1 Development of wood carbon pools (1a) and realized wood harvest (1b) forced by three different RCP scenarios, a present-day (pd) climate, and subject to the harvesting rules of the representative concentration pathways (RCP2.6, RCP4.5 and RCP8.5) and sustained yield (SY2.6, SY4.5, SY8.5, and SYpd). Lines are smoothed over 10 years.**

[Figure]

**S1-Figure 2 Allocation of wood harvest applying sustained yield under present-day climate to different forest regions summed over the entire simulated period (2006-2100).**

[Figure]

[Figure]

---

## Author Comment (AC2) · 17 May 2018

Anonymous Referee #2 Received and published: 23 February 201

Forests are one of Earth's most important resources and wood harvest is one of the main ways in which humans are managing and impacting forest ecosystems. This paper examines the question of how an alternative future wood harvest scenario, based on a "sustained yield" (SY) approach, and responsive to a changing environment, would differ from the standard demand-based wood harvest scenarios currently used by Integrated Assessment Models (IAMs). Under the sustained yield approach, annual wood harvest rates are "optimized" and are equal to the annual rate of re-growth, thus keeping forest stocks maintained at their current levels. In addition, unlike most IAM wood harvest scenarios, the SY approach in this paper uses regrowth rates that are responsive to changing climate and CO2 concentrations, thus potentially allowing higher wood harvest rates than the IAM wood harvest scenarios that are based on static environmental conditions. This work shows one way in which IAMs and ESMs could improve and strengthen their interactions. The paper describes the results of a set of simulations using the dynamic vegetation model JSBACH, forced with climate data from Earth System Model runs from three different Representative Concentration Pathways (and thus three different climate futures). The paper compares results from these JSBACH simulations using both the SY wood harvest scenario as well as the prescribed wood harvest scenarios from the IAMs.

This is an interesting thought experiment, and yet it is not really presented as such in this paper. In fact, one of my main criticisms with the paper is the way in which the work is framed and motivated. The SY approach is unrealistic and it is not clear why, or how, a society would want to pursue a wood harvest plan that involved harvesting more wood than demanded and doing so by harvesting every patch of forest by the exact amount it would regrow each year. It is only in the Discussion that the authors acknowledge that the SY scenario is not meant as a plausible estimate of future wood harvest, but rather as an estimate for the ecological potential for wood harvest. There is also not a lot of explanation given for why IAM demand-based wood harvest scenarios are inherently problematic (aside from not responding to changing environmental conditions). The motivation for using this particular SY approach appears to be a mix of exploring how changing environmental conditions could alter the amount of wood harvest, and also how additional wood harvest could act as a carbon mitigation method. However, the effect of changing environmental conditions on the amount of wood harvested is mixed with the effect of the additional wood harvest that is imposed by the SY approach (which attempts to harvest much more than the demand-based scenarios). In addition, the mitigation potential of the SY approach seems like a fairly inefficient way to capture and store carbon from biomass. The authors also do not address negative impacts of the SY scenario such as the impact on biodiversity (even partial removal of forest would have a negative impact on habitat), or the impact on the overall forest health that could result from continued and widespread human management. Due to the SY approach involving changing forest biomass in response to changing environmental conditions, as well as additional wood harvest to meet the forest regrowth rates, it is difficult to tell how much of the increased wood harvest is coming from the additional forest growth due to changing climate and CO2 concentrations, and how much is coming from choosing to harvest more wood, and in more locations, than the IAM wood harvest scenarios. It would actually be quite interesting to look at this some more and I think the paper would benefit from an additional experiment that was devised to do this (as outlined below).

Response: We gratefully acknowledge the valuable evaluation of the reviewer, who finds the study "an interesting thought experiment", and the useful comments to improve the readability and quality of the manuscript. We revised the manuscript accordingly to all points made by the reviewer and, below, we outline the main specific improvements requested by reviewer #2:

I think the paper would be much improved if a couple of key changes were made: 1) A better framing and motivation for the paper in the Introduction, to make it clear that the SY approach is a thought experiment to examine how much sustainable future wood harvest is possible, and why demand-based wood harvest scenarios are not sufficient.

Key changes 1): Reviewer #2 correctly emphasizes that the motivation for accomplishing this simulation study is to realize the potential of SY in harvesting wood and the long-term effects of wood products on the global carbon cycle for mitigation CO2. We have now used the keyword "potential" in the Introduction and call the study a thought experiment as suggested by the reviewer to avoid misunderstandings. We outline in the discussion now the alternative approaches for the simulation of global forest resource management beyond SY. We discuss the shortcomings and the effects of applying harvesting rules like SY on the ecosystem services besides timber production.

2) In addition to the standard demand-based scenario, and the SY scenario, include a third wood harvest scenario that uses the prescribed demand-based wood harvest scenarios, but allows for forest regrowth rates to change due to changing environmental conditions, and thus allows for changes in the actual wood removed from forests. For example, this scenario could use the prescribed wood harvest area in each gridcell (instead of the wood harvest biomass), or it could use the ratio of prescribed wood harvest biomass to prescribed available forest biomass in each grid-cell. Either of these alternative wood harvest scenarios would retain most of the information from the prescribed demand-based scenario, but would allow the actual biomass harvested to change with changing environmental conditions. This could enable a simple quantification of the impacts of changing climate and $CO_2$ concentrations on future wood harvest, and would separate that effect from the effect of harvesting much more wood under the SY scenario.

Key changes 2): The suggestion of reviewer 2 for a third harvest scenario is an interesting suggestion how to interpret harvest rates prescribed by IAMs in a way such that changing environmental conditions are considered to a certain degree. However, in our study we do not target to show how IAM harvest rates might be interpreted in such a way, but to make an assessment of potentials and to state that these potentials reach far beyond the rates prescribed by IAMs. Regarding the separation of the impact of changing climate and $CO_2$ concentrations and the effect of harvesting regrowth we extended our discussion: We added another simulation to the supplementary material in which we simulate SY harvest under present-day climate. We now refer to this additional scenario in our discussion of the effects of changing climate and $CO_2$ on the achieved harvest rates. We interpret the differences between this SY and SY under RCPS as the effects of changes in $CO_2$ and climate. Additionally, we point out that the differences in harvest potential of the three RCP-SY simulations are solely determined by their different climate and $CO_2$ forcings. Finally, we shortly reflect

upon the differences in harvest amount between the SY simulations and RCPs in the first year of the simulation, which highlight the differences of applying supply side based harvesting (SY) versus demand side harvest (as applied in IAMs) under current climate and $CO_2$ conditions.

Please also note the supplement to this comment:
https://www.biogeosciences-discuss.net/bg-2017-531/bg-2017-531-AC2-supplement.pdf

**Supplement:**

[revised manuscript text omitted]

S1: Supplementary material 1

A1. JSBACH simulations

Simulations were conducted with revision 7277 of cosmos-landveg-fom, a svn branch of revision 7215 of cosmos-landveg, the JSBACH development branch of the department "The Land in the Earth System" of the Max Planck Institute for Meteorology.

Simulations were executed on the IBM Power 6 machine BLIZZARD at the German Climate Computing Center (DKRZ).

Sustained-yield (SY) forest harvesting is implemented in this model version as described in the methods section of the main text. A modification over earlier JSBACH versions is that wood harvest applies just to harvesting from all woody PFTs and specifically from the above-ground wood carbon pool. To isolate the effects of different wood harvest rules, we do not apply land-cover change and dynamic biogeographic vegetation shifts for our future scenarios. We take into account changes in wood carbon pool, natural mortality and forest disturbances to determine the net annual increment of the above-ground wood carbon pool as the maximum amount to be harvested from forest areas.

A1.1 Initial state in 2006

All simulations described in the paper started in 2006 from the same initial conditions. These conditions base on a spin-up of the terrestrial system state using the MPI-ESM climate from the historical (1850-2005) CMIP5 experiment (Giorgetta et al., 2013) and land-use change and wood harvest data from Hurtt et al. (2011).

The initial state was derived carrying out three consecutive simulations. (I) An initial simulation with JSBACH to spin-up photosynthesis, phenology, hydrology and running climatic means required by the disturbance module of JSBACH. This simulation was forced by cycling the first years (1850-1879) of the historical CMIP5 experiment for four times. Wood harvest was fixed to the level of the initial year 1850 and no land-use change was applied. (II) A simulation with the stand-alone carbon cycle module of JSBACH to equilibrate the carbon pools with respect to the driving climate. This simulation was forced by NPP, LAI and climatic means, resulting from the preceding JSBACH simulation. (III) A second JSBACH simulation resuming the first JSBACH simulation, but starting from the equilibrated carbon pools. In this second simulation the full transient (1850-2005) climate from the historical CMIP5 experiment was used and land-use change and wood harvest were prescribed according to the data from Hurtt et al. (2011).

A1.2 Reference level for SY

An important decision for our study is the definition of the reference level of the wood carbon pool to be kept constant in the future applying SY. As one of the main goals of our study is to estimate potentials for wood harvesting under future climate scenarios, consistent with the historic past, we refer to the current level of wood carbon pools. The reference level for the SY simulations was therefore derived from the maximum simulated wood carbon per grid-cell and PFT in the period from 1996 to 2005 under the historical JSBACH simulation (see A1.1 simulation III). Because the historical JSBACH simulation was subject to land-use and land-cover changes maximum wood carbon densities were used instead of wood carbon stocks.

A1.3 Simulation of SY under present-day climate

This simulation (SYpd) keeps the current level of wood carbon pools constant as described above in A1.2. However, it is not forced by a transient but a cycled detrended present-day climate of the period 1991-2020 with a constant CO2 concentration (381 ppm) as the average value of the period (1991-2020). S1-Figure 1 shows the development of wood carbon pool and harvested amount resulting in the simulation SYpd compared to the 6 simulations described in the main text. SYpd realizes a higher wood harvest (+3.2 PgC) than RCPs (~1.2 PgC) at the beginning of simulations and equals the RCP8.5 wood harvest at the end of century. SYpd diverges from wood harvest amount by SY2.6, SY4.5, and SY8.5 largely towards the end of the century and remains below these figures. The geographical allocation of realized wood harvest amount as shown below in SYpd in S1-Figure 2 resembles largely the other SYs (see Figure 2 in the manuscript), however, the amount of harvested wood is lower.

**S1-Figure 1 Development of wood carbon pools (1a) and realized wood harvest (1b) forced by three different RCP scenarios, a present-day (pd) climate, and subject to the harvesting rules of the representative concentration pathways (RCP2.6, RCP4.5 and RCP8.5) and sustained yield (SY2.6, SY4.5, SY8.5, and SYpd). Lines are smoothed over 10 years.**

[Figure]

**S1-Figure 2 Allocation of wood harvest applying sustained yield under present-day climate to different forest regions summed over the entire simulated period (2006-2100).**

[Figure]

[Figure]

---

## Referee Report (RR1)

This is my second time reviewing this manuscript and this version of the manuscript is a significant improvement upon the previous one. The authors have addressed almost all of my previous concerns and suggestions. The motivation for this study is now much clearer, the additional simulations (present day climate scenarios, managed forests, etc) give a lot more detail to the study, and the discussion is much more complete. I particularly liked the inclusion of the "managed forest" mask, which restricts the growth-based harvest method to forest locations that are considered managed - which gives a more realistic bound on the potential of growth-based harvest methods.

I recommend this paper for publication.

---

## Editor Decision (ED1)

[revised manuscript text omitted]

---

## Author Response (AR2)

Report #1

Submitted on 12 Jun 2018
Anonymous Referee #1

**Anonymous during peer-review: Yes** No
**Anonymous in acknowledgements of published article: Yes** No

**Recommendation to the editor**

| | |
|---|---|
| **1) Scientific significance**
Does the manuscript represent a substantial contribution to scientific progress within the scope of this journal (substantial new concepts, ideas, methods, or data)? | Excellent **Good** Fair Poor |
| **2) Scientific quality**
Are the scientific approach and applied methods valid? Are the results discussed in an appropriate and balanced way (consideration of related work, including appropriate references)? | Excellent **Good** Fair Poor |
| **3) Presentation quality**
Are the scientific results and conclusions presented in a clear, concise, and well structured way (number and quality of figures/tables, appropriate use of English language)? | Excellent **Good** Fair Poor |

For final publication, the manuscript should be
**accepted as is**
accepted subject to **technical corrections**
accepted subject to **minor revisions**
**reconsidered after major revisions**
    I am willing to review the revised paper.
    I am **not** willing to review the revised paper.
**rejected**

**Suggestions for revision or reasons for rejection** **(will be published if the paper is accepted for final publication)**

I am grateful to the authors for considering my comments by extending the discussion and making more clear the limitations of their modelling. However, still it is very misleading to impose a concept about "sustained yield" (SY) by estimating wood harvest rates, which are perceived as providing a potential, but which have nothing to do with the actually possible harvest. An information about a harvest when harvesting is hardly possible (e.g. because of inaccessibility) or even impossible provides actually not a potential. This should better not be converted into carbon in harvested wood pools. A harvest potential is usually something which may potenially be used in future, but which is currently not used for some reasons. A potential SY is thus possibly not what the authors actually reveal with their simulations, but rather a kind of a change rate in biomass stocks. It would be much better to avoid SY. If harvesting is to be implemented in these models, and this would certainly be a good idea, it makes only sense if the harvesting regime would at least show some realism.

*R1: We thank the reviewer for his/her additional comments, which made it clear to us that the terminology and framing of our paper can lead to confusion in particular when considering the interdisciplinary context of our study. We thank the editor for a chance to modify the manuscript accordingly. The following major changes were implemented:*

1- *We outline in the introduction in detail the different concepts dealt with in our study and how they relate to each other and we have clarified our terminology throughout the manuscript. Specifically we now distinguish between a "growth potential" and a "harvest potential", where the second accounts for the reviewer's and editor's concern that much area may not be suited for management (see next bullet point), while the first is now clarified as a purely ecological potential. Accordingly, we rename the applied harvest concept to "growth-based harvest (GB)" to clearly state the relation between forest growth potential under changing climate and $CO_2$ concentration and the amount of wood harvest as regrown annually. We discuss the similarity to the sustained yield concept (targeting the annual increment), but refrain from mixing this management practice with our approach throughout the rest of the manuscript.*

2- *We provide new figures masking out inaccessible forest area by overlaying a map of managed forest area to truly account for real potentials of forest harvest management for mitigating $CO_2$ (MF). This map indicats forest areas subject to conservation, infrastructural limits, or not being influenced by human activities so far due to other reasons by Kraxner et al., 2017. The resulting numbers of harvest and mitigation potentials are lower than our estimates based on growth potential. The impact of environmental changes on harvest potentials, however, turns out to remain important. The key message of our paper, that simulating wood harvest needs to account for these changes, remains valid qualitatively, but becomes stronger as it can now better be linked to considerations of actual future harvest regimes, as suggested by the review.*

3- *We extended substantially the discussion of the trustworthiness of our model with respect to processes like $CO_2$-fertilization as suggested by the editor. In particular, we now included the findings of a recent study that used a similar model version like ours but included the nitrogen cycle explicitly (it found a low sensitivity of the land carbon cycle to nitrogen limitation). We also point the reader to new evidence that $CO_2$-fertilization of our model may be in line with observations. We further added new simulation results of forcing GB and MF by present-day climate to show the effects of transient climate and $CO_2$ concentrations on forest growth and harvest potential. This makes it easier for the reader to identify the effect of climate changes in all components, from growth potential, to harvest potential, to relevance for mitigation.*

*These changes have substantially improved the quality of our submitted (see differences below). We hope this response satisfies the high standards of Biogeosciences.*

[revised manuscript text omitted]

Sum of harvest/growth potential 2006-2100 $[kgCm^{-2}]$

**Figure 3**

[Figure]

[Figure]

[Figure]

**Figure 4**

[Figure]

[Figure]

S1: Supplementary material 1

A1. JSBACH simulations

Simulations were conducted with revision 7277 of cosmos-landveg-fom, a svn branch of revision 7215 of cosmos-landveg, the former JSBACH development branch of the department "The Land in the Earth System" of the Max Planck Institute for Meteorology.

Simulations were executed on the IBM Power 6 machine BLIZZARD at the German Climate Computing Center (DKRZ).

Growth-based (GB) forest harvesting is implemented in this model version as described in the methods section of the main text. A modification over earlier JSBACH versions is that wood harvest applies just to harvesting from all woody PFTs and specifically from the above-ground wood carbon pool. To isolate the effects of different wood harvest rules, we do not apply land-cover change and dynamic biogeographic vegetation shifts for our future scenarios. We take into account changes in wood carbon pool, natural mortality and forest disturbances to determine the net annual increment of the above-ground wood carbon pool as the maximum amount to be harvested from forest areas.

A1.1 Initial state in 2006

All simulations described in the paper started in 2006 from the same initial conditions. These conditions base on a spin-up of the terrestrial system state using the MPI-ESM climate from the historical (1850-2005) CMIP5 experiment (Giorgetta et al., 2013) and land-use change and wood harvest data from Hurtt et al. (2011).

The initial state was derived carrying out three consecutive simulations. (I) An initial simulation with JSBACH to spin-up photosynthesis, phenology, hydrology and running climatic means required by the disturbance module of JSBACH. This simulation was forced by cycling the first years (1850-1879) of the historical CMIP5 experiment for four times. Wood harvest was fixed to the level of the initial year 1850 and no land-use change was applied. (II) A simulation with the stand-alone carbon cycle module of JSBACH to equilibrate the carbon pools with respect to the driving climate. This simulation was forced by NPP, LAI and climatic means, resulting from the preceding JSBACH simulation. (III) A second JSBACH simulation resuming the first JSBACH simulation, but starting from the equilibrated carbon pools. In this second simulation the full transient (1850-2005) climate from the historical CMIP5 experiment was used and land-use change and wood harvest were prescribed according to the data from Hurtt et al. (2011).

A1.2 Reference level for GB

An important decision for our study is the definition of the reference level of the wood carbon pool to be kept constant in the future applying GB. As one of the main goals of our study is to estimate potentials for wood harvesting under future climate scenarios, consistent with the historic past, we refer to the current level of wood carbon pools. The reference level for the GB simulations was therefore derived from the maximum simulated wood carbon per grid-cell and PFT in the period from 1996 to 2005 under the historical JSBACH simulation (see A1.1 simulation III). Because the historical JSBACH simulation was subject to land-use and land-cover changes maximum wood carbon densities were used instead of wood carbon stocks.

A1.3 Simulation of GB and MF under present-day climate

These simulations (GBpd and MFpd) keep the current level of wood carbon pools constant as described above in A1.2 and apply the GB harvest rule to global and managed forest area. However, they are not forced by a transient but a cycled detrended present-day climate of the period 1991-2020 with a constant $CO_2$ concentration (381 ppm) as the average value of the period (1991-2020). S1-Figure 1 shows the development of wood carbon pool and harvested amount resulting in the simulation GBpd compared to the 6 simulations described in the main text. GBpd realizes a higher wood harvest (+3.2 PgC) than RCPs (~1.2 PgC) at the beginning of the simulations and equals the RCP8.5 wood harvest at the end of the century. GBpd diverges from the GB2.6, GB4.5, and GB8.5 wood harvest amounts largely towards the end of the century and remains below these figures. The geographical allocation of realized wood harvest amount as shown below in GBpd in S1-Figure 2 resembles largely the other GBs (see Figure 2 in the manuscript), however, the amount of harvested wood is lower. Values for the simulated wood harvest from MFpd is lower than GBpd because of limiting forest harvest to managed forest area (excluding primary forest area). S1-Figure 3 shows the net mitigation of MF forced by present day climate. Logically, the annual harvest amount stay more or less constant (~3.2 PgC) in the 21st century. This is exactly resembling the concept of sustained yield if no changes in forest growth is expected. As a result, MFpd would result in a lower net mitigation potential (~150 PgC) than GB and MF (see Figure 4 in text for details), applying the same life cycle analysis described in section 2.6.

**S1-Figure 1 Development of global standing wood carbon pools forced by three different RCP scenarios and a present-day (pd) forcing, subject to the harvesting rules of the representative concentration pathways (RCP2.6, RCP4.5 and RCP8.5) or subject to growth-based harvesting (GB2.6, GB4.5, GB8.5, and GBpd) (1a). Development of RCP wood harvest rates, of the growth potential of forests under GB and of the harvest potential under GB limited to global managed forest area (MF2.6, MF4.5, MF8.5, and MFpd) (1b). All lines are smoothed over 10 years.**

[Figure]

[Figure]

**S1-Figure 2** Allocation of wood harvest applying growth-based harvesting rule to the global forest area (GB) and limited to managed forest area (MF) under present-day forcing summed over the entire simulated period (2006-2100).

[Figure]

[Figure]

References¶
Giorgetta MA et al. (2013) Climate and carbon cycle changes from 1850 to 2100 in MPI-ESM simulations for the Coupled Model Intercomparison Project phase 5. J. Adv. Model. Earth Syst. 5:572–597.¶
Hurtt, GC et al (2011) Harmonization of land-use scenarios for the period 1500–2100. 600 years of global gridded annual land-use transitions, wood harvest, and resulting secondary lands. Climatic Change 109:117–161.

**S1-Figure 3 Net mitigation potentials of simulated wood harvest from growth-based harvest rule applied to managed forest area under cycled present day forcing (MFpd). Left axis shows the annual carbon fluxes due to harvested material and product decay changing atmospheric CO$_2$ concentration, and the mitigation potential of wood products as the difference of both. Right axis accumulates the annual figures over time.**

[Figure]

---

## Author Response (AR3)

Chair of Forestry Economics and Forest Planning

• Tennenbacher Str. 4 • D-79106 Freiburg

[Figure]

Editorial Board of
Biogeosciences

**Faculty of Environment and Natural Resources**

**Dr. Rasoul Yousefpour**

Telefon 0761 / 203-36 88

Telefax 0761 / 203-36 90

E-mail: rasoul.yousefpour@ife.uni-freiburg.de

http://www.ife.uni-freiburg.de

Datum: 16.11.2018

**Dear Editor,**
**Dear Prof. Dr. Anja Rammig,**

We thank you for your effort with our submitted paper in discussion, bg-2017-531 "Simulating sustained yield harvesting adaptive to future climate change".

We appreciate the reviews contributing to improvement of our manuscript. However, we would like to raise attention to the report from review round #3 forwarded to us on 2018/11/06.

Mainly, we perceive the main argument of the reviewer as unfounded. Report-1 of review round #3 states that our concept design *"implies that any forest worldwide is regarded as a potential wood harvest, no biodiversity hotspot, conservation areas or last wilderness areas are excluded from the potential area, thus carbon potential for wood harvest. **The only restriction applied refers to accessibility."***

This is wrong. We describe clearly our approach on MF (managed forests) in section 2.5, e.g. lines 168-170: "we conduct a post-processing step overlaying a map that masks out forest areas subject to conservation, infrastructural limits, or not being influenced by human activities so far due to other reasons". We continue by explaining that we used for this purpose the published map of managed forests produced by Kraxner et al., 2017.

This change in our methodology, which we introduced following the second round of revisions, satisfied the former reviewer of the manuscript (report 2) fully as she/he clearly states "I particularly liked the inclusion of the "managed forest" mask, which restricts the growth-based harvest method to forest locations that are considered managed - which gives a more realistic bound on the potential of growth-based harvest methods."

Beyond the main criticism of review round #3 being unfounded we have substantial problems with understanding the comments raised by report-1:

- Sentences are not complete e.g. *"The Life-cycle analysis is limited to the decay rates of the 3 reason why wood is harvested, bioenergy, paper and wood products."*

- Statements are scientifically incorrect or subjective, e.g**.,** *„Thus, by design this study has a very global view on forest*

*conditions, neglecting that natural forests, i.e. woods, are not managed and should not be managed, i.e. harvested. It seems to be triggered by the image of European forestry of what forests are worldwide that I find highly disputable, it ignores biome- or ecoregion- specific conditions.*"

- The review asks for information that exists in the manuscript, e.g.,
    - *"An overview on the methodological approach of computing wood harvest in IAMs has to be provided in order to allow the reader to compare and follow on the carbon estimates of wood products quantified by ESM (your study) and IAMs (the approach you suggest to improve). There are several IAMs being used in the scenario-production work, but you do not cite any of them to substantiate your argument that the climate-dependent simulation of wood harvest is considered."* and again „*Lines 308-310: statement not substantiated by methods and clear explanation of the criticized IAM approach, no citation to IAM publication provided.*" → We explain and reference three IAM approaches in l. 356-367.
    - „*The introduction section should end with a clear explanation of the modelling concept of this study (the abstraction of representing respective carbon fluxes and pools) and its objectives to provide the reader with an overview of what to expect and to cross-check later on what to conclude from this study.*" → l. 95-103 explains the goals of this study, l. 103-112 gives a general summary of the approach to assessing mitigation potentials, l. 66-94 gives a general summary of the growth simulation in our model, ESMs and IAMs in general. Carbon fluxes and pools specifically in our model are detailed in section 2.1
    - „*Lines 405-407: experiments and setting not explained in methods",* while there is the reference in l. 407 to the supplemental text S1, which explains the experimental setup.
- The following comment (and similar comment later) reveals a lack of understanding of some fundamentals of carbon cycle science: „*Line 108-111: This assumption [using an impulse response function] is flawed because it ignores carbon emissions from industry, fossil fuels and land-use change. […] Such an assumption implies that only GPP and respiration fluxes are exchanged between the ocean and terrestrial vegetation, and that carbon release from*

[Figure]

*wood products is the only additional source of carbon. […] This assumption needs to be revised as it is central to the entire study."* An impulse response function approach approximates the uptake of emissions by natural sinks in land/ocean and is a common tool to estimate the fraction of emissions held by the atmosphere at year x after the emission occurred (e.g., Caldeira and Kasting, 1993; Pongratz et al., 2011; O'Halloran et al., 2012; Millar et al., 2017). It makes no statements about the source of emissions (wood harvest, fossil fuel, …) as the $CO_2$ uptake in the sinks is independent of the source of carbon. It is only important that the response curves are derived under comparable climate/$CO_2$ conditions. Even if the reviewer's criticism of our method were valid -- the analysis of the mitigation potential would only be a side aspect of our study that does not affect our main conclusion and is *not* *"central to the entire study".*

This list deals with the major comments by the reviewer, but even more comments exist in the review that we perceive as inappropriate for the standards of a scientific journal. The overall picture to us is that the reviewer is not capable or not willing of assessing the quality of our study. There is no reason to expect he/she would be capable or willing to perform an adequate review in a further round of revisions. We therefore ask you, as the editor, to evaluate our manuscript ignoring the unfounded comments by review round #3. Considering the editor's expertise in the field of climate-vegetation-interactions and the fact that our manuscript has been seen by two reviewers before (with the one reviewer, who re-evaluated our response (report 2), recommending that our revised version should be published), we believe the review process has been covering a large amount of external expertise already in the prior rounds of review. Therefore, we are willing to take into account any final comments the editor finds crucial and with that finalize the paper for publication.

Best regards,

Rasoul Yousefpour on behalf of all co-authors

References:

Caldeira, K., and Kasting, J.F., Insensitivity of global warming potentials to carbon dioxide emission scenarios. Nature 366, 1993.

O'Halloran, T., Law, B.E., et al., Radiative forcing of natural forest disturbances. Global Change Biology 18, 2012.

Pongratz, J., Caldeira, K., Reick, C.H., and Claussen, M., Coupled climate−carbon simulations indicate minor global effects of wars and epidemics on atmospheric $CO_2$ between AD 800 and 1850. The Holocene 21(5), 2011.

Millar, A. J., Niclolls, Z. R., Friedlingstein, P., Allen, M. R., A modified impulse-response representation of the global response to carbon dioxide emissions. Atmospheric Chemistry and Physics 17, 7213-7228, 2017

[Figure]

UNI FREIBURG

---

## Author Response (AR4)

Chair of Forestry Economics and Forest Planning

• Tennenbacher Str. 4 • D-79106 Freiburg

[Figure]

Editorial Board of
Biogeosciences

**Faculty of Environment and Natural Resources**

**Dr. Rasoul Yousefpour**

Telefon 0761 / 203-36 88

Telefax 0761 / 203-36 90

E-mail: rasoul.yousefpour@ife.uni-freiburg.de

http://www.ife.uni-freiburg.de

**Dear Editor,**
**Dear Prof. Dr. Anja Rammig,**

We thank you for your effort with our submitted paper in discussion, bg-2017-531 "Simulating growth-based harvest adaptive to future climate change".

We are fully grateful for the constructive comments raised by the editor and the opportunity to finalize the manuscript for the publication in advanced journal of Biogeosciences.

Datum: 19.12.2018

Best regards,

Rasoul Yousefpour on behalf of all co-authors

---

## Author Response (AR5)

Chair of Forestry Economics and Forest Planning

• Tennenbacher Str. 4 • D-79106 Freiburg

[Figure]

Editorial Board of
Biogeosciences

**Faculty of Environment and Natural Resources**

**Dear Editor,**
**Dear Prof. Dr. Anja Rammig,**

**Dr. Rasoul Yousefpour**

Telefon 0761 / 203-36 88

Telefax 0761 / 203-36 90

E-mail: rasoul.yousefpour@ife.uni-freiburg.de

http://www.ife.uni-freiburg.de

We thank you for your effort with our submitted paper in discussion, bg-2017-531 "Simulating growth-based harvest adaptive to future climate change".

We have now finalized the manuscript for the publication in advanced journal of Biogeosciences.

Best regards,

Datum: 30.12.2018

Rasoul Yousefpour on behalf of all co-authors